# Cross-scale causal information flow from El Niño Southern Oscillation to precipitation in eastern China

Yasir Latif[1], Kaiyu Fan[2,3], Geli Wang[2,*], and Milan Paluš[1,*]

[1]Department of Complex Systems, Institute of Computer Science of the Czech Academy of Sciences, 182 00 Prague 8, Czech Republic
[2]Key Laboratory for Middle Atmosphere and Global Environment Observation, Institute of Atmospheric Physics, Chinese Academy of Sciences, Beijing 100029, China
[3]Dalian Meteorological Bureau, Dalian 116001, China
[*]Joint corresponding authors

**Correspondence:** Geli Wang (wgl@mail.iap.ac.cn) and Milan Paluš (mp@cs.cas.cz)

**Abstract.** The El Niño/Southern Oscillation (ENSO) is a dominant mode of climate variability influencing temperature and precipitation in distant parts of the world. Traditionally, the ENSO influence is assessed considering its amplitude. Focusing on its quasi-oscillatory dynamics comprising multiple time scales, we analyze causal influence of phases of ENSO oscillatory components on scales of precipitation variability in eastern China, using information-theoretic generalization of Granger causality. We uncover the causal influence of the ENSO quasi-biennial component on the precipitation variability on and around the annual scale, while the amplitude of the precipitation quasi-biennial component is influenced by the low-frequency ENSO components with the periods around 6 years. This cross-scale causal information flow is important mainly in the Yellow River basin, while in the Yangtze River basin the causal effect of the ENSO amplitude is dominant. The presented results suggest that in different regions different aspects of ENSO dynamics should be employed for prediction of precipitation.

## 1 Introduction

The Asian summer monsoon (ASM) is the most active monsoon system on the planet, bringing ample moisture from the tropical ocean to the continent and accounting for more than half of yearly rainfall (Krishnamurti, 1971; Wang and LinHo, 2002). The ASM is divided into two major sub-systems: the South Asian summer monsoon (SASM), which is characterized by a pronounced low-level westerly wind, and the East Asian summer monsoon (EASM), characterized by a pronounced low-level southerly wind (Wang et al., 2003). Recent climate change and related extreme weather events leading to catastrophic life and property loss in the East Asian region, particularly in China, necessitate water resource management and security measures which must take into account substantial regional and intra-seasonal fluctuation in precipitation (Heng et al., 2020). The Yangtze River Basin (YZRB) has complicated and unusual precipitation patterns, a unique regional climate, and is a flood-prone area due to the influences of the East Asian and South Asian summer monsoons (Huang et al., 2021; Sutcliffe, 1987). The EASM, which is caused by a heat difference between the Pacific Ocean and the Asian mainland, is a significant component of inland East Asia's climate. The regional precipitation pattern in China, in particular, is significantly correlated

with the EASM (Li et al., 2023b). Precipitation is the primary factor influencing agricultural and economic development in YZRB (Lijuan et al., 2018).

The El Niño/Southern Oscillation (ENSO) (McPhaden et al., 2006) is rooted in complex nonlinear large-scale interactions within and between the atmosphere and ocean circulation, and causes a persistent abnormal fluctuation in sea surface temperature (SST) in the central and eastern equatorial Pacific. This fluctuation has a quasi-periodic character with a two- to seven-year period (Wang, 2018). In the beginning, the notion of El Niño (EN) and Southern Oscillations (SO) originated by figuring out an ocean-atmosphere positive feedback that triggers ENSO (Bjerknes, 1969). The initial warm SST anomaly in the equatorial eastern Pacific reduces the east-west SST gradient and slows the Walker circulation, culminating in the westerly wind anomaly in the equatorial central Pacific (Gill, 1980). The westerly wind anomaly, in turn, promotes a change in ocean circulation, exacerbating the SST anomaly. Positive feedback causes the tropical Pacific to warm, culminating in El Niño. Once El Niño develops, negative feedbacks are required to transition from a warm to a cold phase, a process called La Niña (Wingfield et al., 2018). Precipitation varies inextricably with internal and external oscillations in global sea surface temperature. The Pacific Ocean's alternate cycle of warming (El Niño) and cooling (La Niña) states effects interannual climate variability (Pui et al., 2012; Webster and Yang, 1992). ENSO impacts global climate through its teleconnections that may serve as a reliable source of predictability (Horel and Wallace, 1981), however, are highly sensitive to global temperature changes (Tsonis et al., 2003; Philip and van Oldenborgh, 2006; Zheng et al., 2016) . Climate change poses a severe threat to China's water security, as extreme precipitation events become more frequent, such as high precipitation events in Beijing in 2012, and more recently in Henan in 2021, which resulted in billions of dollars in economic losses (Mingzhong et al., 2017; Li et al., 2019; Xie et al., 2015; Hsu et al., 2022). The long-term patterns in EASM rainfall in the Yellow River Basin (YWRB) were driven by variations in air circulation over the Pacific Ocean, such as the ENSO cycle and the Pacific Decadal Oscillation (PDO) phase transition, which may thrive as measurable factors in the prediction of future EASM rainfall (Li et al., 2023a). The links between extreme precipitation and ENSO depend on magnitude, regions, and seasons (Wei et al., 2012).

**Table 1. Short literature review**. Previous studies regarding the impact of ENSO on annual and seasonal precipitation in Chinese regions. * Et: Evapotranspiration, Ppt: Precipitation, YWRB: Yellow River Basin, YZRB: Yangtze River Basin, ERB: East River Basin

| Authors | Region/study period | Variable | Observation |
|---|---|---|---|
| Yang et al., 2004 | YWRB* (1951-2000) | Ppt* and Et* | Decreased Ppt and increased Et during 1990-2000 |
| Li and Zheng., 2013 | YWRB (1951-2012) | Ppt and ENSO | Decadal weakening of autumn Ppt due to ENSO |
| Zhang et al., 2013 | ERB* (1956-2005) | Ppt and ENSO | Strong correlation between ENSO and April Ppt |
| Xiao et al., 2015 | YZRB* (1960-2019) | Ppt and ENSO | Strong relationship between ENSO and seasonal Ppt |
| Zhang et al., 2016 | YZRB (1979-2015) | Ppt and ENSO | Dominant/predictable impact of ENSO on Asian Ppt |
| Gao and Wang., 2017 | YWRB (1960-2011) | Extreme Ppt | Weakening of summer monsoon |
| Cao et al., 2017 | YZRB (1960-2015) | Ppt and ENSO | Strong ENSO impact on wetting and drying Ppt pattern |
| Chang et al., 2017 | YWRB (1956-2010) | Ppt and runoff | Abrupt change in Ppt with insignificant trends at 8 stations |
| Hardiman et al., 2018 | YZRB (1992-2015) | Ppt and ENSO | Linear impact of ENSO on summer Ppt |
| Lv et al., 2019 | China (1960-2013) | Ppt and ENSO | Decreased Ppt but increased extreme events attributed to ENSO |
| Liu et al., 2020 | YWRB (1961-2017) | Seasonal Ppt | Linear impact of ENSO on winter and spring Ppt |

Several studies (Yang et al., 2005; Li and Zeng; Zhang et al., 2013; Xiao et al., 2015; Zhang et al., 2017; Gao and Wang, 2017; Cao et al., 2017; Chang et al., 2016; Hardiman et al., 2018; Lv et al., 2019; Liu et al., 2020) have previously investigated the association between annual/seasonal precipitation variability and ENSO in the Yangtze and Yellow River Basins of China. The current impact of ENSO on precipitation in various sub-basin to basin scales across China and main river basins (Yangtze and Yellow) is summarized in Table 1. In addition to the aforementioned studies, other recent studies considered combined influence of ENSO and NAO or PDO and future ENSO projections (Qadimi et al., 2021; Alizadeh, 2022; Liu et al., 2023a). Some studies proposed strengthening of ENSO events (Cai et al., 2021) in the 21st century due to global warming, while others expect ENSO weakening (Callahan et al., 2021).

Considering quasi-cyclic character of ENSO, Jajcay et al. (2018) studied interactions between different quasi-periodic components of ENSO dynamics. Interactions among the oscillatory ENSO components and the role of triadic resonances and synchronization in the explanation of super El-Niños were analysed by Pires and Hannachi (2021).

Jajcay et al. (2018) uncovered a complex causal network involving instantaneous phases and amplitudes of annual, quasi-biennial and low-frequency (period 4–7 years) ENSO modes. The observed causal interactions lead to intermittent synchronization phenomena responsible for extreme ENSO events. In this study we analyze causal influence of instantaneous phases of ENSO oscillatory components on scales of precipitation variability in eastern China, using information-theoretic generalization of Granger causality. Previous studies were restricted to SST amplitude-based ENSO states and their influential role in large scale interactions and precipitation variability. The term "ENSO phases" is frequently used for the three cases of high-amplitude positive ENSO+ (El Niño); high-amplitude negative ENSO- (La Niña), or low-amplitude, neutral ENSO0 case. We will use here the term "ENSO states" in order to avoid confusion with the instantaneous phases of ENSO oscillatory modes.

In the following sections we will describe the study area and the analyzed data. Then we will introduce the applied methods, from the scale-wise decomposition using the complex continuous wavelet transform (CCWT thereafter), through the conditional mutual information (CMI) as the casuality measure and surrogate data method for assessing its statistical significance, to the conditional means defined as tools for estimating the effect of the uncovered causal relations in measurable physical quantities. Then we will present the results and their discussion.

## 2 Data and methods

### 2.1 Study area

The present study includes a particular area of Chinese region through which major Chinese rivers; Yangtze and Yellow flow and ultimately drains into sea as shown in Fig. 1. The Yangtze and Yellow Rivers have distinct natural habitats and development demands, despite the fact that they both originate on the Qinghai-Tibet Plateau (Fang et al., 2021). The Yellow River, also known as China's Mother River, initiates in the Bayankala Mountains and travels eastward throughout the Loess Plateau (LP) and the North China Plain eventually draining into the Bohai Sea. In accordance with 1973 survey, the length of the Yellow River is 5,464 kilometers, and its basin area of 752,443 km$^2$ consists of three primary sub-basins: The Tibetan Plateau (TP), the LP, and the alluvial plateau in the east (Fu et al., 2004). The altitudes range from 2,000 to 5,000 m in the Tibetan Plateau in

the western sections, which stretch from the Bayankala Mountains to the eastern estuary, and from 500 to 2,000 m in the LP and alluvial plateau in the east.

The Yangtze River is China's longest river and the world's third largest, contributes considerably to China's equitable economic and ecological growth (Xiao et al., 2015). The ecological growth of the Yangtze River Basin refers to the development and changes in its ecosystems over time, influenced by both natural processes and human activities. The Yangtze River Basin has undergone significant ecological changes due to various factors such as climate, geography, and human impact. The basin's high, middle, and lower portions have various climates and geomorphology, which contribute to its great biodiversity and huge number of uncommon and unique species (Chen, 2020). Therefore, its ecological growth is as important as the economic growth.

Yangtze River's primary course commences at the TP and travels 6,300 kilometers east to the Eastern China Sea. The YZRB is predominantly controlled by Siberian northwest winter and southeast summer monsoon. This monsoon brings cold, dry air from Siberia during the winter months. It has the potential to diminish temperatures and precipitation, resulting in drought conditions in certain areas of the basin (Yang et al., 2023). Yichang hydrological station (YHS) separates the Yangtze River into upper and lower sections and is renowned as the 'Gateway to the Three Gorges'. The Three Gorges Dam (TGD) lies just approximately 40 kilometers above (Xu et al., 2007). The territory above Yichang station is commonly regarded as the upper sub-basin of the YZRB; the region from Yichang station and Hukou station is the middle sub-basin; and the region under Hukou station is the lower sub-basin of the YZRB (Fang et al., 2018). The YZRB lies in subtropical and temperate climate zone dominated by monsoonal winds; the southern region exhibits subtropical climate while northern region presents temperate zone. Major flooding in YZRB is linked with warm ENSO and strong summer monsoons typically occur after El Niño conditions in the winter, while weak winter monsoons occur after La Niña (Xu et al., 2007). Our area of study covers southeastern part of YZRB.

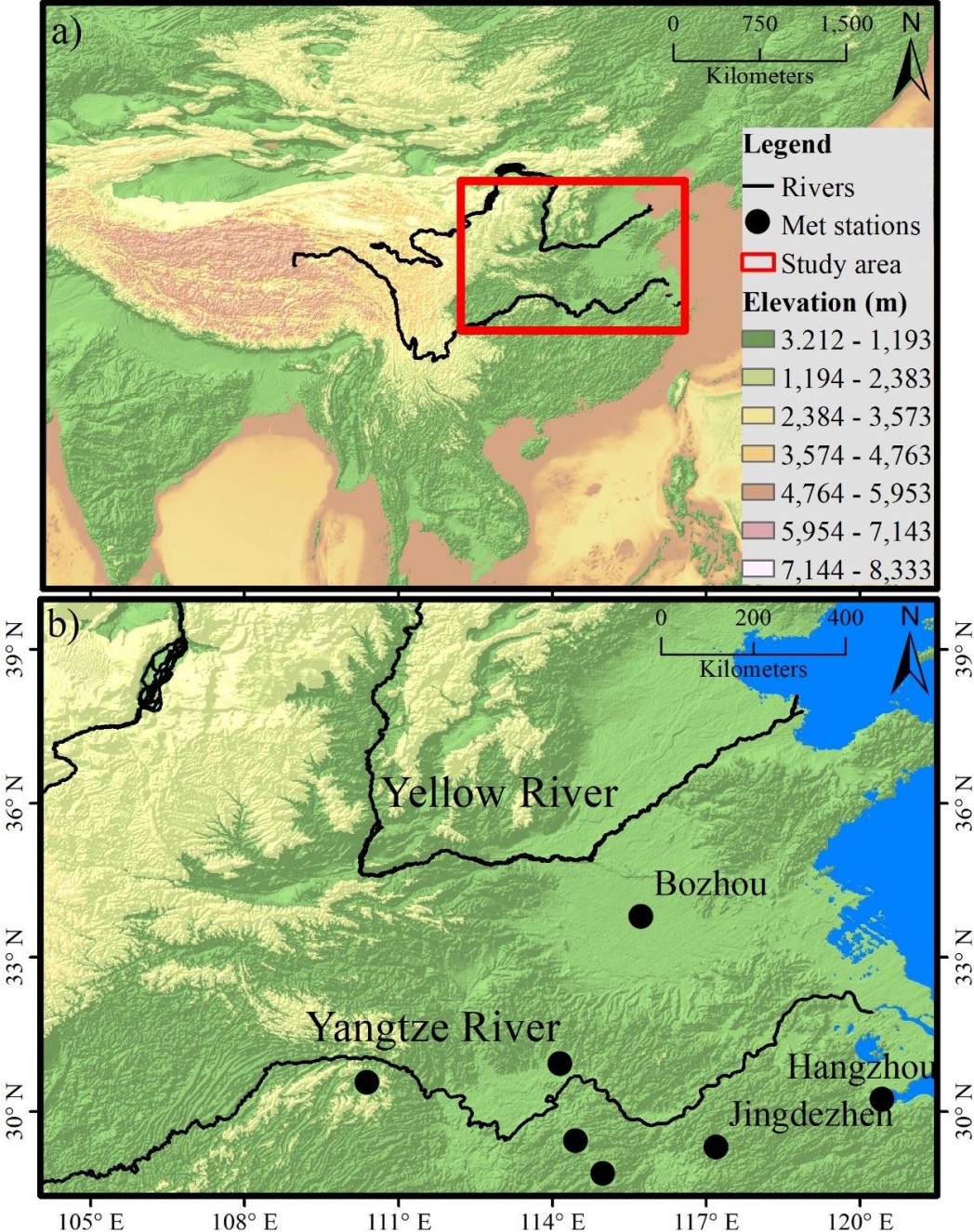

**Figure 1. Study area.** (Top) Localization of the selected region in Yangtze and Yellow River Basins. (Bottom) Study area in a detailed view, including the positions of selected stations.

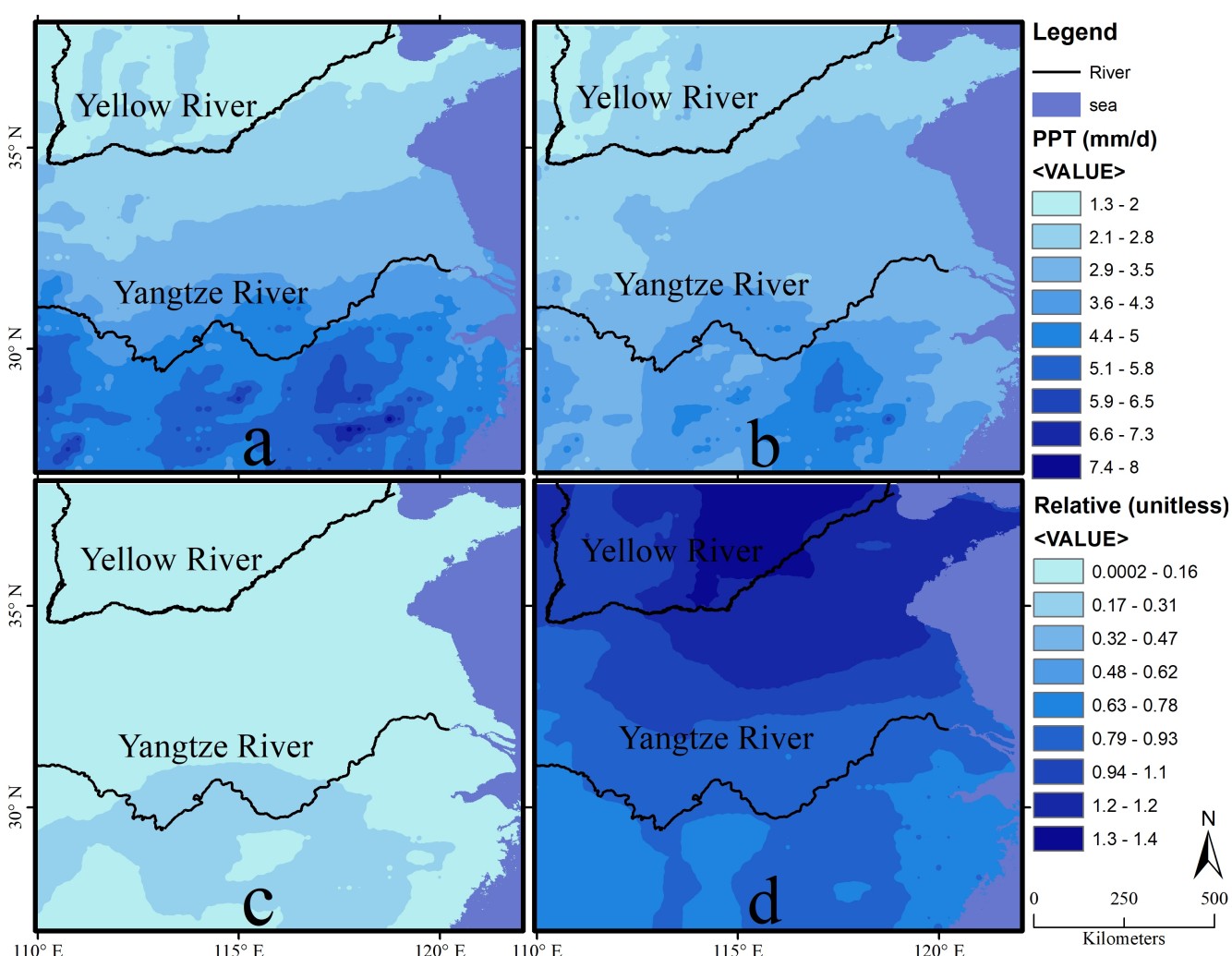

**Figure 2. Precipitation and its variability in the study area.** Spatial distribution of precipitation and its variability during 1951-2020; (a) mean precipitation, (b) precipitation standard deviation (SD), (c) relative difference between ENSO positive and ENSO neutral state, (d) relative precipitation SD (SD/mean precipitation).

## 2.2 Gridded data

We have used fifth-generation atmospheric reanalysis of European Centre for Medium-Range Weather Forecasts (ECMWF), namely ERA 5. The fifth generation was launched in 2017 by the Copernicus Climate Change Service (Jiang et al., 2021). Jiang et al. (2021) have explicitly explained the advantages of ERA 5 concerning advanced assimilation system and parameterization schemes as compared to the previously launched generations. Furthermore, the spatial and temporal range was improvised by enabling the hourly estimation at a horizontal resolution of 0.25° covering 137 vertical levels.The ERA dataset is on 1.5° × 1.5° grids from 1979 to present. The SST over the tropical Pacific is based on the Hadley Centre sea ice and sea surface

temperature data (HadISST) (Rayner et al., 2003). The top level is 0.01 hPa around 80 km above the ground surface. The ERA 5 data is freely available for users. Our data range from 1951 to 2020 for the selected study area and were downloaded from https://cds.climate.copernicus.eu/cdsapp#!/home.

      Recently, some studies focused on comparing the performance of model-based precipitations such as ERA to satellite products for mainland Chinese regions and Tibetan Plateau, since reliable precipitation retrievals with fine spatiotemporal resolu-
tions are vital in global and regional evaluations (Xu et al., 2022; Hu and Yuan, 2021). Model-based precipitation estimates, which are an essential alternative to satellite-based precipitation products, have grown rapidly in recent decades. Model-based products outperform satellite products in subregions of temperate monsoon climate (TM) and temperate continental climate (TC) (Xu et al., 2022). However, when compared to gauge precipitation, ERA 5 performance was being compromised in terms of frequency and intensity for Tibetan Plateau (Hu and Yuan, 2021). The latter study further argued that rainfall gauges on the
Tibetan Plateau are generally positioned in valleys and may not correctly reflect the region's average. Another study for the same regions of TP and Sichuan province observed that ERA-Interim exhibits better performance than IMERG_E, IMERG_L, IMERG_F, CHIRPS, TRMM_3B42, TRMM_3B42RT (Lei et al., 2021). Future study will require additional observations and clarification of station locations and higher levels (Hu and Yuan, 2021). ERA 5 has replaced ERA- Interim and this release offers several improvements over the previous ERA-Interim reanalysis solution due to improved design and generation
methodologies. In comparison to ERA-Interim, this dataset is more advanced due to several factors including a high resolution, day-by-day archiving, diverse data sources, better assimilation, and diversified data products (Tarek et al., 2020). The assessment at the monthly flood season (Lavers et al., 2022) indicates that the ERA5 is slightly better than the other models. It is better in the extratropics. ERA5 precipitation has been found to be a sufficiently excessive source of information in the non-tropical areas. Therefore, it is suggested that ERA5 be utilized primarily for extratropical precipitation monitoring. ERA5
performs spatially across China, with the highest correlation coefficient values in eastern, northwestern, and north China and the lowest biases in southeast China (our study area) (Jiao et al., 2021). Similarly, intensity comparisons show strong agreement between ERA-5 and EOBS in Germany, Ireland, Sweden, and Finland, but some disagreement in places with scarce input stations (Rivoire et al., 2021).

      Fig. 2 illustrates the distribution of mean precipitation in the south of Eastern China from 1951-2020, its standard devia-
tion and relative difference of ENSO states. The mean precipitation is high in the southeastern part of YZRB compared to the YWRB. It reaches its maximum limit of 8 mm/day in some areas of YZRB while it remains in the lower range of 1.4 to 3.5 mm/day in upper areas of YWRB as shown in Fig. 2a. The standard deviation, characterizing the overall precipitation variability, also exhibits the same pattern as the mean precipitation. The variability is higher in southeastern parts of YZRB whereas it is lower for YWRB as shown in Fig. 2b. It is interesting to assess the variability relative to mean precipitation, therefore standard deviation divided by the mean precipitation is shown in Fig. 2d. The relative variability is much higher in
YWRB compared to YZRB. It reaches the maximum level of 1.4 in most of the areas of YWRB, however, remains at 0.63 to 0.93 in the southeastern parts of YZRB. For comparison, we map in Fig. 2c the difference between the ENSO positive and neutral states related to the mean precipitation. We observe that the relative difference is lower than the overall variability in the

entire selected region but increases in the same heavy precipitation southeastern parts of YZRB attaining the range of 0.17-0.31.

## 2.3   Station data and EASM index

The observational data used in this paper are monthly precipitation records (from January 1955 to December 2016) provided by National Meteorological Information Center, China Meteorological Administration (http://data.cma.cn/). Monthly East Asian Summer Monsoon index (from January 1948 to December 2015) is defined by Zhang et al. (2003).

**Table 2.** Geographical coordinates of seven local precipitation stations used in the combined region of Yellow and Yangtze River basins

| Station ID | Station name | Province | Longitude | Latitude |
| --- | --- | --- | --- | --- |
| 57355 | Huangjiawan | Hubei | 110.4 E | 31 N |
| 57494 | Wuhan | Hubei | 114.1 E | 30.6 N |
| 57598 | Hejiadian | Jianxi | 114.6 E | 29 N |
| 57799 | Yankeng | Jiangxi | 114.9 E | 27.1 N |
| 58102 | Bozhou | Anhui | 115.7 E | 33.7 N |
| 58457 | Hangzhou | Zhejiang | 120 E | 30.2 N |
| 58527 | Jingdezhen | Jiangxi | 117.2 E | 29.3 N |

## 145   2.4   ENSO data

Niño3.4 series was downloaded from https://www.cpc.ncep.noaa.gov/data/indices/. ENSO states were defined as warm (ENSO+) and cold (ENSO-) periods based on crossing the threshold of $\pm0.5°C$ for the Oceanic Niño Index (ONI) (3 month running mean of ERSST.v5 SST anomalies in the Niño 3.4 region). Neutral ENSO0 means ONI between $\pm0.5°C$. ONI was obtained from https://origin.cpc.ncep.noaa.gov/products/analysis_monitoring/ensostuff/ONI_v5.php.

## 150   2.5   Scale-wise decomposition, instantaneous phases and amplitudes

Consider a time series $\{x(t)\}, t = 1, 2, 3, \ldots, N$, here either monthly Niño3.4 index or precipitation recordings, reflects dynamics on different time scales. The latter can be extracted, i.e., the time series decomposed using the complex continuous wavelet transform (CCWT thereafter) with the complex Morlet wavelet (Torrence and Compo, 1998)

$$\psi(t) = \frac{1}{\sqrt{2\pi\sigma_t^2}} \exp\left(-\frac{t^2}{2\sigma_t^2}\right) \exp\left(2\pi i f t\right), \qquad 1$$

where $i = \sqrt{-1}$, $\sigma_t$ is the bandwidth parameter, and $f$ is the central frequency of the wavelet. $\sigma_t$ determines the rate of the decay of the Gauss function, its reciprocal value $\sigma_f = 1/\pi\sigma_t$ determines the spectral bandwidth. CCWT converts the time

series $x(t)$ into a set of complex wavelet coefficients $W(t, f)$:

$$W(t,f) = \int\limits_{-\infty}^{\infty} \psi(t')x(t-t')dt'. \qquad 2$$

The central wavelet frequency $f$ defines the related time scale. Due to the limited spectral bandwidth we obtain an oscillatory quasi-periodic component $W(t, f)$ reflecting temporal variability at that time scale. Using analytic signal approach (Pikovsky et al., 2001), a complex oscillatory time series can be represented as

$$W(t,f) = s_f(t) + i\hat{s}_f(t) = A_f(t)e^{i\phi_f(t)}, \qquad 3$$

where $s_f(t) = \text{Real}\{W(t,f)\}$, $\hat{s}_f(t) = \text{Im}\{W(t,f_i)\}$, and

$$\phi_f(t) = \arctan\frac{\hat{s}_f(t)}{s_f(t)}, \qquad 4$$

is the instantaneous phase, and

$$A_f(t) = \sqrt{s_f(t)^2 + \hat{s}_f(t)^2} \qquad 5$$

is the instantaneous amplitude of the oscillatory component $W(t, f)$. Paluš (2014) describes how the instantaneous phases and amplitudes can be used to uncover causal cross-scale information transfer.

## 2.6 Conditional mutual information as a causality measure

Paluš (2014) describes in detail the use of the conditional mutual information as a causality measure for inferring cross-scale causal relations. Here we briefly remind the basic ideas.

Mutual information $I(X;Y)$ of two random variables $X$ and $Y$, is defined as $I(X;Y) = H(X) + H(Y) - H(X,Y)$, where the entropies $H(X)$, $H(Y)$, $H(X,Y)$ are given in the usual Shannonian sense (Cover and Thomas, 1991). The conditional mutual information $I(X;Y|Z)$ of the variables $X$, $Y$ given the variable $Z$ is defined using the conditional entropies (Cover and Thomas, 1991; Paluš, 2014) as

$$I(X;Y|Z) = H(X|Z) + H(Y|Z) - H(X,Y|Z). \qquad 6$$

Consider two time series $\{x(t)\}$ and $\{y(t)\}$ regarded as realizations of two stationary ergodic stochastic processes $\{X(t)\}$ and $\{Y(t)\}$ which represent observables of two possibly coupled systems. Alternatively, the time series $\{x(t)\}$ and $\{y(t)\}$ can be understood as one-dimensional projections of trajectories of dynamical systems $\dot{\mathbf{X}} = f_X(\mathbf{X},\mathbf{Y})$ and $\dot{\mathbf{Y}} = f_Y(\mathbf{Y},\mathbf{X})$, where $\mathbf{X}$ and $\mathbf{Y}$ are vectors of dimensions $d_1$ and $d_2$, respectively.

Paluš et al. (2001) proposed to measure the information transferred from system (process) $\{Y(t)\}$ to system (process) $\{X(t)\}$ using the conditional mutual information $I(\mathbf{Y}; \mathbf{X}_\tau | \mathbf{X})$, where $\mathbf{X} = \mathbf{X}(t)$ and $\mathbf{X}_\tau = \mathbf{X}(t + \tau)$.

Using the idea of Markov processes, Schreiber (2000) introduced a functional of conditional probability distributions called transfer entropy. Paluš and Vejmelka (2007) show that the transfer entropy is equivalent to CMI $I(\mathbf{X}; \mathbf{Y}_\tau | \mathbf{Y})$. Barnett et al.
(2009) have shown analytically that the transfer entropy (i.e., CMI $I(\mathbf{X}; \mathbf{Y}_\tau | \mathbf{Y})$) is equivalent to Granger causality for Gaussian processes. Therefore, causal influence is frequently interpreted as information transfer, or directed information flow. However, it is worth noting that this interpretation use of the term "information flow" might not be compatible with the term meaning in information physics (Perdigão et al., 2020; Hall and Perdigão, 2021) or in theory of dynamical systems where the information flow can be derived from system equations (Liang, 2013).

If the measurement of information about the future $X_\tau$ of the process $\{X\}$, shifted $\tau$ time units forward ("$\tau$-future" thereafter), contained in the process $\{Y\}$ is used for testing the existence of a causal link from $\{Y\}$ to $\{X\}$, denoted as $Y \rightarrow X$, Paluš and Vejmelka (2007) show that the vectors $\mathbf{X}$ and $\mathbf{Y}_\tau$ can be substituted by one-dimensional components $x$ and $y_\tau$, and the CMI in the time series representation reads as

$$I(y(t); x(t + \tau) | x(t), x(t - \eta_1), \dots x(t - (d_1 - 1)\eta_1)). \qquad 7$$

The condition in CMI (7) must contain complete information about the state of the system $X$ (Paluš and Vejmelka, 2007). According to the theorem of Takens (1981), the state of a $d_1$-dimensional dynamical system (a point in the state space) is mapped by the set of time-lagged coordinates $x(t), x(t - \eta_1), \dots x(t - (d_1 - 1)\eta_1)$, where $\eta_1$ is the backward time-lag used in the embedding of system $X$. This time-lag can be set according to the embedding construction recipe based on the first minimum of the mutual information (Fraser and Swinney, 1986).

The causal link $X \rightarrow Y$ is tested in analogy with (7):

$$I(x(t); y(t + \tau) | y(t), y(t - \eta_2), \dots y(t - (d_2 - 1)\eta_2)). \qquad 8$$

For estimating the information transfer delay, Wibral et al. (2013) proposed the following CMI reformulation:

$$I(x(t); y(t + \tau) | y(t + \tau - 1), y(t + \tau - 1 - \eta_2), \dots y(t + \tau - 1 - (d_2 - 1)\eta_2)), \qquad 9$$

in which the condition moves forward with the increasing prediction horizon $\tau$, while in the formulation of Paluš and Vejmelka
(2007) the condition is kept in the same position for all values of $\tau$. The Wibral et al. (2013) formula 9 is used in order to establish the causal delay, while the formulas 7 and 8 are used for testing the statistical significance of uncovered causal relations. Wibral et al. (2013) formula 9 was proposed as a CMI in the case of Self Prediction Optimality (SPO) of $y$ states prior to the forecast delay $\tau$. This is a very conservative estimate of CMI/TE since the SPO may be never reached with CMI/TE of Eq. 9 being underestimated. CMI estimated according to Eq. 7 or 8 is more sensitive with respect to the detection of causality.

For testing the cross-scale causality, before applying Eq. 8, the Niño3.4 and the precipitation data underwent CCWT and $x(t)$ is substituted by the ENSO phase $\phi_{f_i}(t)$ for a particular frequency $f_i$, and $y(t)$ by the precipitation amplitude $A_{f_j}(t)$ for a frequency $f_j$. The Gaussian estimator was used and $d_2 = 3$ was chosen as in (Paluš, 2014) based on "saturation of the results", i.e., obtaining unchanged results for $d_2 = 4$ in comparison with $d_2 = 3$. The tested value is the CMI average for time lags $\tau = $ 1 to 6 months, according to the recommendation in (Paluš and Vejmelka, 2007).

## 2.7    Surrogate data for statistical testing

Finite-sample estimates of mutual information are always nonzero. In order to assess the presence of causal relations in the analyzed data it is suitable to relate the CMI values computed from studied data to ranges of CMI values obtained from uncoupled processes that share statistical properties of the analyzed data. Using the surrogate data testing procedure we manipulate the original data in a randomization procedure that preserves the original frequency spectra or variance on all relevant time scales. In this study we use the circular time-shifted surrogate data, proved effective for the inference of causality (Manshour et al., 2021). For the analyzed time series $X$ of the length $N$, we generate 100 independent realizations of time-shifted surrogates as follows: For each realization an integer variable $k < N$ is randomly chosen. Then, by moving the first $k$ values of $X(1), X(2) \ldots X(k)$ to the end of the time series, we generate the circular time-shifted surrogate series $X^{surr}$ as

$$X^{surr} = \{X(k+1), X(k+2), \ldots, X(N), X(1), X(2), \ldots, X(k)\}. \qquad 10$$

In order to avoid surrogates very close to the original series, or an influence of seasonality, $k$ is constrained as follows: $\min(k, N - k) > 200$ and $3 < \text{mod}(k, 12) < 10$. The results of the surrogate data tests can be represented as the $Z$-score, e.g., for CMI, marked as $I$, it is

$$Z = \frac{I_d - \overline{I_s}}{\sigma_s}, \qquad 11$$

where $I_d$ is the CMI value estimated from the studied data, $\overline{I_s}$ is the mean for 100 realizations of the surrogate data and $\sigma_s^2$ is 230   their variance. Typically, the results are considered statistically significant for $Z > 2$. In the cross-scale analyses, the surrogate data are applied directly to the raw data before the application of the wavelet transform, using different shifts for the phase and amplitude series.

## 2.8    Conditional means as the effect size

The results reported below present a statistical evidence of the cross-scale causal influence of ENSO on precipitation variability 235   in eastern China. In order to quantify this causal effect in a measurable physical quantity, and compare it with the effect of the amplitude-based ENSO states we employed the method of conditional means (CM) (Jajcay et al., 2016), illustrated in Fig. 3. A segment of Niño3.4 time-series data is shown by black colour in Fig. 3a, while the ENSO states are marked by colouring: light red is used for El Niño, i.e., the ENSO positive state, light blue for La Niña, i.e., the ENSO negative state, and white for

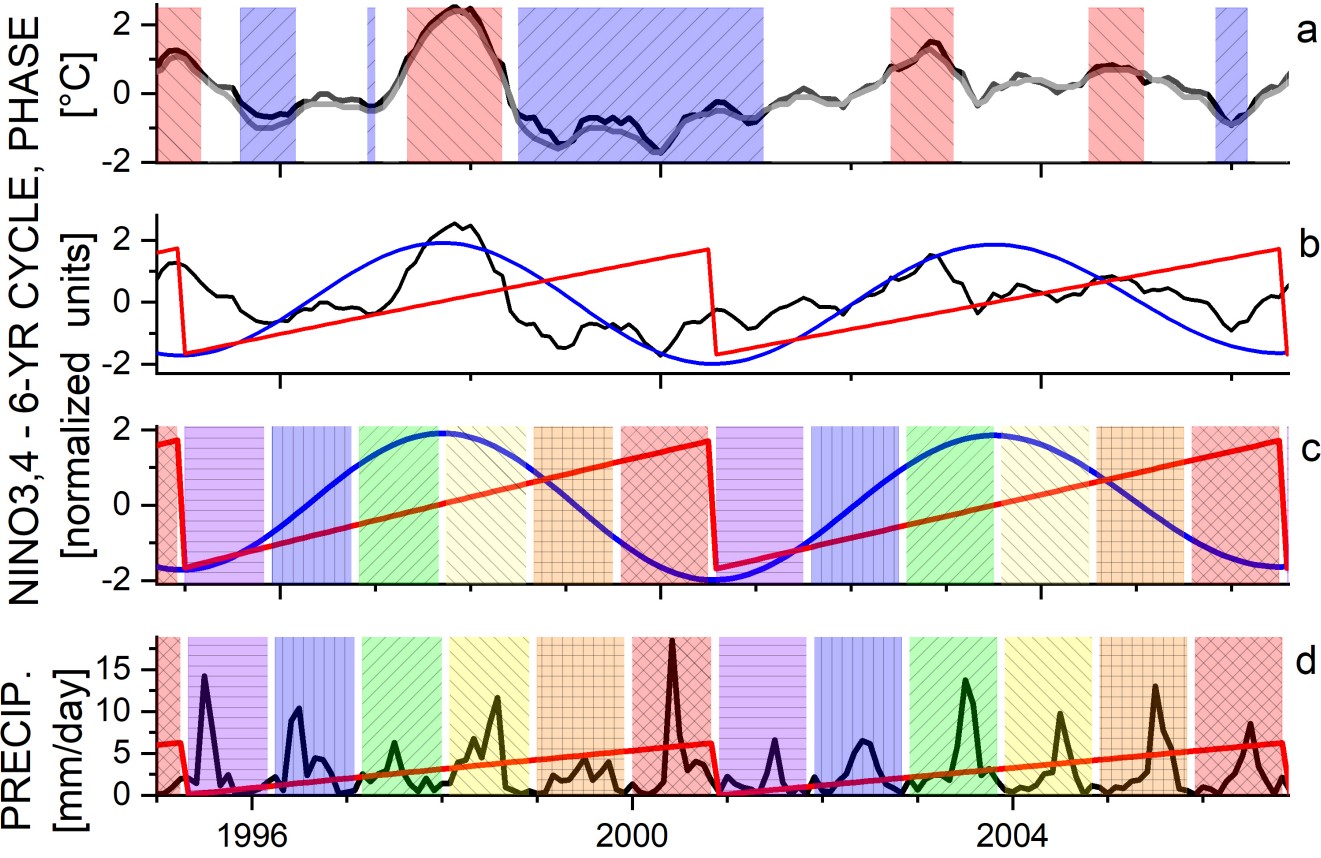

**Figure 3. ENSO states and binning of the low-frequency cycle.** From top to bottom: (a) A segment of anomalized Niño3.4 (black) and ONI (gray) time series with marked ENSO states: warm episodes ENSO+ (light red), cold episodes ENSO- (light blue) and neutral ENSO0 state (white). (b) The same segment of anomalized Niño3.4 time series (black) with its CCWT-extracted 6-yr component (blue) and the instantaneous phase (red) of the latter. (c) The 6-yr Niño3.4 component (blue) and its instantaneous phase (red). The bars of different colors and patterns mark the 6 phase bins into which each 6-yr cycle is divided. (d) A segment of reanalysis precipitation data from the gridpoint 33.75°N 115.75°E (black) and the 4 months lagged phase (red) of the 6-yr Niño3.4 cycle and related phase bins (bars of different colors and patterns) in which the precipitation conditional means are computed.

the ENSO neutral state given by the Oceanic Niño index between $-0.5$ and $0.5$. The Niño3.4 time series (black) is plotted

again in Fig. 3b together with its CCWT extracted 6-yr oscillatory component (blue curve) and the instantaneous phase (red

saw-like pattern) of the latter. In each cycle, the phase rises from $-\pi$ to $\pi$ (note that the data in Fig. 3b are normalized for

the common scale plot), however, the angular range of one cycle ($2\pi$) is equal to six years only approximately, i.e., the cycle

period is variable within a small range given by the CCWT bandwith. Each cycle of $2\pi$ radians is divided into six equidistant

bins (different colours and patterns in Fig. 3c, d). Thus each bin is only approximately equal to one year in the real time. The

phase bins are used to compute precipitation conditional means in order to see patterns of precipitation variability related to the

considered ENSO cycle. The conditional mean for a particular bin is obtained by averaging precipitation data (from a particular

station or a grid point) belonging to that bin in all cycles, e.g., the averaging runs over all blue bins over the whole dataset (see Fig. 3d for a subset of the data).

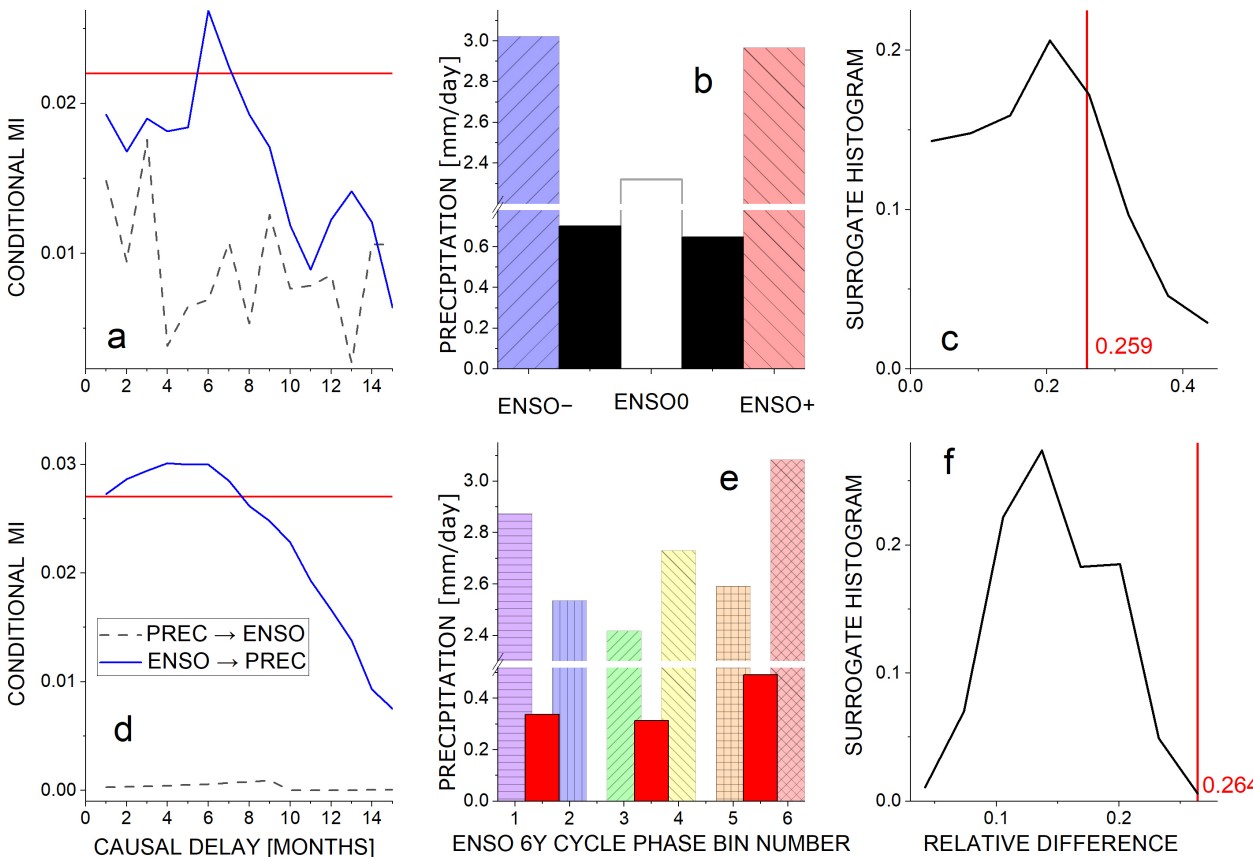

**Figure 4. Causal mechanisms and their effects.** (a) Conditional mutual information measuring the causal influence of ENSO states on precipitation characterized by the EASMI-ZQY index (solid blue line) and causality in the opposite direction (dashed black line). The red line is the significance threshold given as the mean+2SD for the surrogate data. (b) Conditional means for the precipitation data from the gridpoint 33.75°N 115.75°E for different ENSO states (ENSO- light-blue, ENSO0 white, ENSO+ light-red) computed for the lag of 6 months. Differences of the adjacent states in black. (c) Evaluation of statistical significance of the maximum relative difference between states, here ENSO- and ENSO0 (red vertical line) using the histogram for the surrogate data (black), (d) Conditional mutual information measuring the causal influence of ENSO 6yr cycle phase on 2yr cycle amplitude for precipitation characterized by the EASMI-ZQY index (solid blue line) and causality in the opposite direction (dashed black line). The red line is the significance threshold given as the mean+2SD for the surrogate data. (e) Conditional means for the precipitation data from the gridpoint 33.75°N 115.75°E for the 6 phase bins within the ENSO 6yr cycle (various colors). Differences of adjacent bins (red) considered as the amplitude of the precipitation quasi-biennial cycle. The effect of the 6yr cycle phase is estimated as the maximum difference of the bin values – here the difference between the values of the 6th (orange) and the 3rd (yellow) bins. This value relative to the total precipitation mean is 0.264 and is marked by red vertical line in (f) and found statistically significant in comparison with the surrogate histogram (black).

Computing the conditional means, the precipitation time series is not exactly aligned in time with the ENSO states or the ENSO phase bins, since the causal effect of ENSO can occur with some time delay. The causal delay can be found in the causality analysis as follows: In Fig. 4a the conditional mutual information represents the causal influence of ENSO states on

the precipitation (EASMI-ZQY index). It was computed using the Wibral et al. (2013) formula 9 in which $x(t)$ is a discrete 3-valued function of the ENSO states, $y(t)$ is the precipitation EASMI-ZQY index discretized into four bins using the equiquantal binning algorithm (Paluš and Vejmelka, 2007), $d_2 = 1$. It is plotted as a function of causal delay. The blue colour shows the causal influence in the direction from ENSO to precipitation while the dashed black line shows the causal influence of the precipitation on the ENSO states. The significance level is shown by the red line. It is evident that there is no significant causality from the precipitation to the ENSO, however, the influence of ENSO exhibits a significant peak for the time lag of 6 months. Therefore, computing the conditional precipitation means for the ENSO states, precipitation data are advanced by 6 months.

Fig. 4b represents the conditional means, i.e., the average precipitation for the ENSO states using the precipitation data at the grid point (33.7° N 115.75° E). The three ENSO states, i.e., negative, positive and neutral, have been shown by light-blue, light-red and white bars, respectively. The differences between the two adjacent states have been marked by the black bars. The first black bar represents the difference between negative and neutral ENSO states, while the next one exhibits the difference between the positive and neutral ENSO states. The maximum difference, expressed as a relative value (the difference divided by the average precipitation) is equal to 0.259 which is shown by the vertical red line in Fig. 4c, while the black line illustrates the histogram of the same differences obtained from the surrogate data. It can be observed that the red bar lies inside the surrogate histogram which means the difference between two ENSO states is not statistically significant in this grid point.

Fig. 4d shows the conditional mutual information showing the causal influence of the phase of the 6-year component obtained from the Niño3.4 time series, on the precipitation amplitude for the variability in the quasi-biennial scale (blue). It was again computed using the Wibral et al. (2013) formula 9, however, since the cross-scale causality is evaluated, before applying Eq. 9, the Niño3.4 and the precipitation data underwent CCWT and $x(t)$ is substituted by the ENSO phase $\phi_{f_i}(t)$ for a particular frequency $f_i$, and $y(t)$ by the precipitation amplitude $A_{f_j}(t)$ for a frequency $f_j$. The Gaussian estimator was used and $d_2 = 3$ was chosen as in (Paluš, 2014). It is evident again that there is no significant causality from the precipitation to the ENSO phase shown by the black dashed line. However, the influence of the ENSO phase on the amplitude of precipitation exhibits a clear significant peak in approximately four months (lags 2-6 months). Therefore, for the calculation of conditional means for the six phase bins, we used the ENSO phase bins having the time shift of four months back relative to precipitation data. Fig. 4e represents the results of conditional means computed in different ENSO phase bins, marked by different light colours, while differences between the adjacent bins are displayed as red bars.

In order to evaluate the influence of the low-frequency ENSO mode on the precipitation, the difference is taken between the maximum and minimum of the conditional means in the six phase bins. Here, again expressed as the relative value, it is 0.264 and is illustrated by the vertical red line in Fig. 4f. The histogram obtained from the surrogate data using the same procedure shows this value outside the surrogate distribution. Thus the effect of the slow ENSO cycle on precipitation in this grid point is statistically significant.

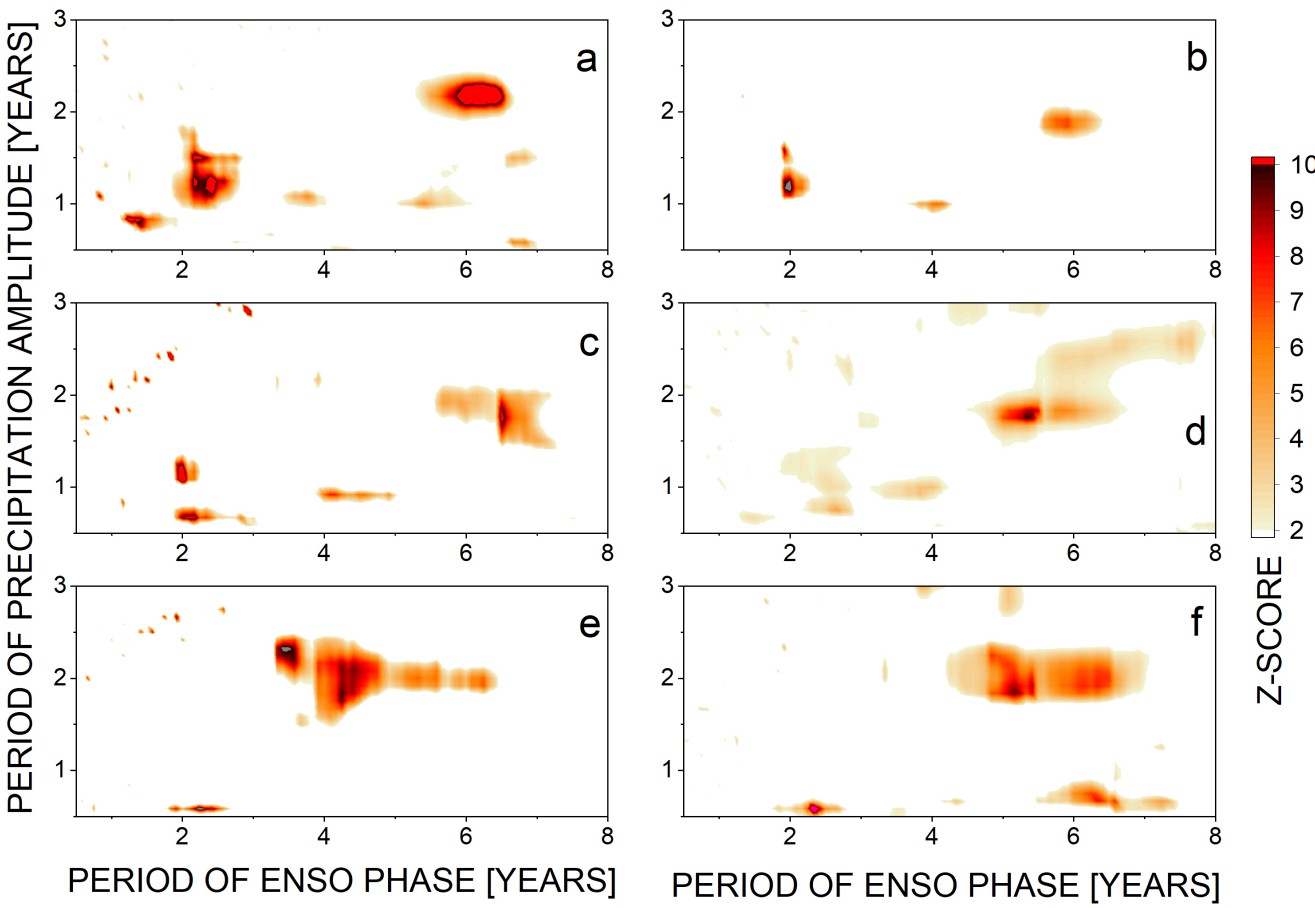

**Figure 5. Cross-scale ENSO influence on precipitation in eastern China.** Cross-scale phase-amplitude information transfer characterizing the causal influence of the phase of ENSO oscillatory components, with periods given on the abscissa, on the amplitude of precipitation oscillatory components with periods given on the ordinate. Significant causal influence of ENSO detected in (a) EASMI-ZQY index, (b) precipitation data from 6 stations from Hu Bei, Jiang Xi and Zhejiang provinces (see Tab. 2, averaged results), (c) precipitation data from station 58457 Hangzhou from Zhejiang province, (d) precipitation data from station 58527 Jingdezhen from Jiangxi province, (e) precipitation data from 58102 Bozhou station from An Hui province, (f) ERA 5 reanalysis precipitation data from the gridpoint 33.75°N 115.75°E. The colour codes present the conditional mutual information $Z$-score for $Z > 2$, obtained in the test using 100 realizations of surrogate data.

## 3   Results

285   The cross-scale causal influence of ENSO on eastern China precipitation, measured as the information transfer from the time series of instantaneous phase of an oscillatory ENSO component to the time series of instantaneous amplitude of a precipitation oscillatory component was evaluated using conditional mutual information and surrogate data testing approach (see Methods) and is presented in Fig. 5. The ENSO phase for a particular time scale was extracted from Niño3.4 index using complex continuous wavelet transform. Similarly, the precipitation amplitude for a particular time scale was also extracted using the

290   CCWT for relevant central period from the EASMI-ZQY index, from station precipitation data, or reanalysis data. First, we

evaluate the causal ENSO effect on the Eastern Asian Summer Monsoon index characterizing the whole eastern China region – the EASMI-ZQY index (Zhang et al., 2003) (Fig. 5a). The dominant patterns of statistically significant causality have been identified as an influence of the phase of the ENSO oscillatory component with the period close to two years (quasi-biennial, or QB component thereafter) on the the amplitude of the annual cycle in the precipitation index and related variability on the time scale close to one year (quasi-annual or QA variability thereafter). The QA precipitation variability is also influenced by slower ENSO oscillations with periods between 4 and 6 years. Another identified area of the cross-scale phase-amplitude causality is the influence of the slow ENSO modes (periods starting under 5 and ending over 6 years) on the biennial and quasi-biennial precipitation variability. The results from the EASMI-ZQY index are, in various extents, repeated in individual station precipitation data. For instance, in the precipitation data from Hangzhou station 58457 from Zhejiang province (Fig. 5c) we can see some influence of the ENSO QB and slow modes on the QA precipitation variability and a marked influence of the phase of the low-frequency ENSO modes (periods between 5 and 7 years) on the amplitude of QB precipitation variability. In the precipitation data from Bozhou station 58102 from An Hui province (Fig. 5e) the amplitude of the precipitation QB variability is influenced by a broadband low-frequency (LF) ENSO oscillatory mode with periods in the range from 3 to 7 years. In all figures (but Fig. 5b) we can see a number of small spots of false positive results which occur due to multiplicity of the tests in the phase period × amplitude period plane. This effect can be partially attenuated by taking an average over results from several stations such as the results in Fig. 5b. The only spots of significant causality which survived the averaging give the influence of the ENSO QB mode on the precipitation QA variability and the influence of the ENSO low-frequency mode with the period around 6 years on the amplitude of the QB precipitation variability.

The reported results present a statistical evidence of the cross-scale causal influence of ENSO on precipitation variability in eastern China. In order to quantify this causal effect and compare it with the effect of the amplitude-based ENSO states we employed the method of conditional means (CM) (Jajcay et al., 2016), illustrated in Fig. 3.

The conditional means of precipitation, conditioned either on the ENSO states or on the 6 phase bins, derived from the instantaneous phase of the low-frequency ENSO component with the period around 6 years, were computed and their maximum differences were statistically evaluated for all grid points in the selected area and mapped in Fig. 6. Thus Fig. 6 illustrates two different approaches to measure the causal effect of ENSO on precipitation in the South Eastern Chinese region. Fig. 6a represents the map of the maximum relative difference (RD) in the 6 phase bins, while Fig. 6b shows the maximum RD between ENSO states (positive, negative and neutral). In order to see the causal effects in physical quantities, Fig. 6 c and d represent absolute values for the same variables as selected in a and b.

In Fig. 6b, d, we evaluate the influence of ENSO states, i.e., from the point of view of ENSO oscillatory dynamics it is the influence of the ENSO amplitude on precipitation. The value of RD are high and statistically significant in the south of YZRB (Fig. 6b, d), while in the case of the influence of the phase of low-frequency modes of ENSO, the significant areas are located mainly around YWRB (Fig. 6a, c).

Extending the circle of investigation, we further observed the values and the occurrence of maxima and minima of the precipitation conditional means in relation to either ENSO states, or to the 6 phase bins of the low-frequency ENSO component phase. Considering the ENSO states, the precipitation conditional means maxima, expressed relatively to the total mean, peak

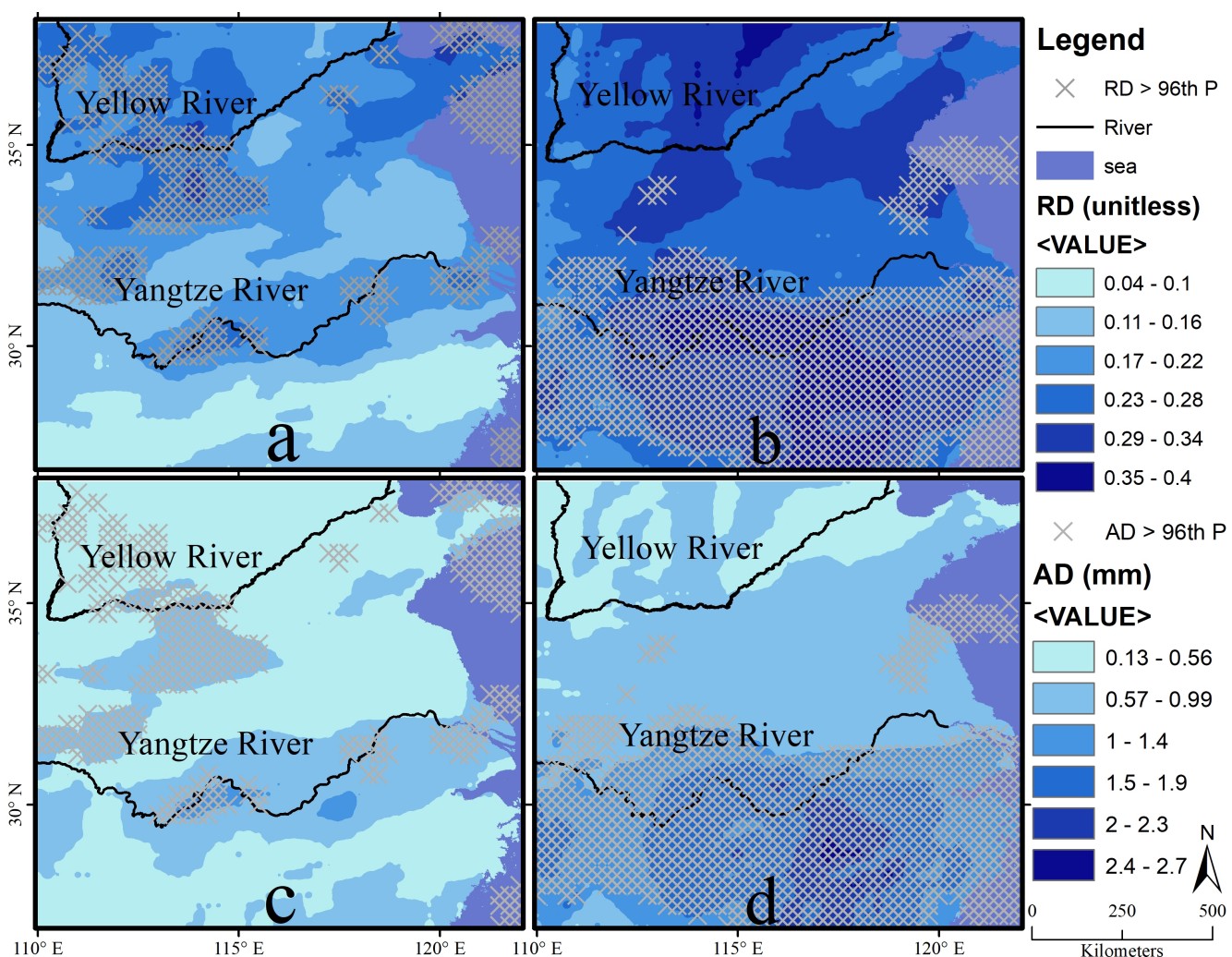

**Figure 6. Quantification of the effects of two causal mechanisms.** Relative (a, b) and absolute (c, d) maximum differences of precipitation conditional means: (a, c) conditioning on the six phase bins, i.e., the effect of the phase of the low-frequency ENSO component on precipitation; and (b, d) conditioning on the three ENSO states, i.e., the effect of the ENSO amplitude on precipitation. Statistically significant differences marked by X.

in YZRB with ranges from 1.08 to 1.23 (Fig. 7a), while the minima range from 0.8 to 0.9 for the entire study area but increase at a few locations of the lower YZRB (Fig. 7b). Answering the question in which ENSO state the precipitation conditional means maxima occur, the ENSO positive state dominates, mainly in YZRB, while in YWRB the maxima occur in the ENSO negative state (Fig. 7c). The precipitation conditional means minima occur exclusively in the ENSO neutral state (Fig. 7d).

Considering the 6 phase bins of the ENSO low-frequency component and the precipitation conditional means maxima, expressed relatively to the total mean, the highest values ranging from 1.1 to 1.23 dominate the YWRB region. Only a few areas in YZRB reach this range, while the most of the YZRB study area receives values from 1.03 to 1.09 (Fig. 8a). This

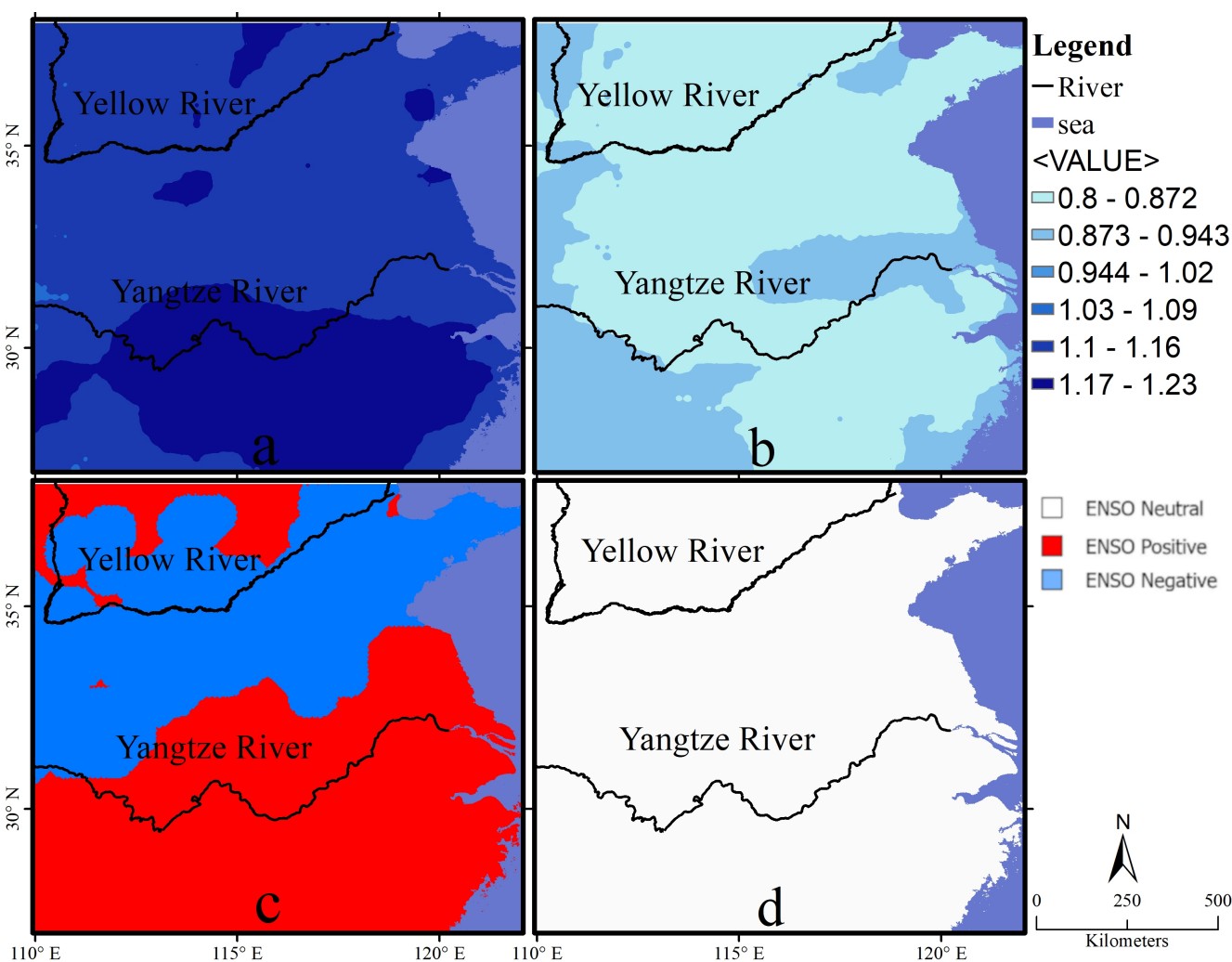

**Figure 7. Geografy of ENSO amplitude influence.** Maximum and minimum precipitation values in ENSO states; (a) maximum precipitation value, (b) minimum precipitation value, (c) ENSO state in which maximum precipitation occurs, (d) ENSO state in which minimum precipitation occurs.

is another evidence, in addition to the areas of statistically significant differences in Fig. 6a, c, that in the YWRB area the dominant ENSO causal mechanism is due to the phase of the low-frequency ENSO component and not the ENSO amplitude.

The precipitation conditional means minima peak in the lower reaches of YZRB ranging from 0.96 to 1.02, while the lowest values, under 0.89 are confined in the north-western quadrant of the study area. In the majority of the study area the values range from 0.89 to 0.95 (Fig. 8b). Discussing the occurrence of CM extrema within the LF cycle, we can refer to bin numbers, or divide the cycle into three states, using the term negative state for bins 1 and 6 at the edge of cycle reaching cycle minima, the positive state for central bins 3 and 4 where the cycle peaks, and neutral state for bins 2 and 5 (cf. Fig. 3c). In YZRB and

southward from it, the minima occur in phase bins 2, 5 and 6 (Fig. 8d), that is in neutral and negative states of the low-frequency

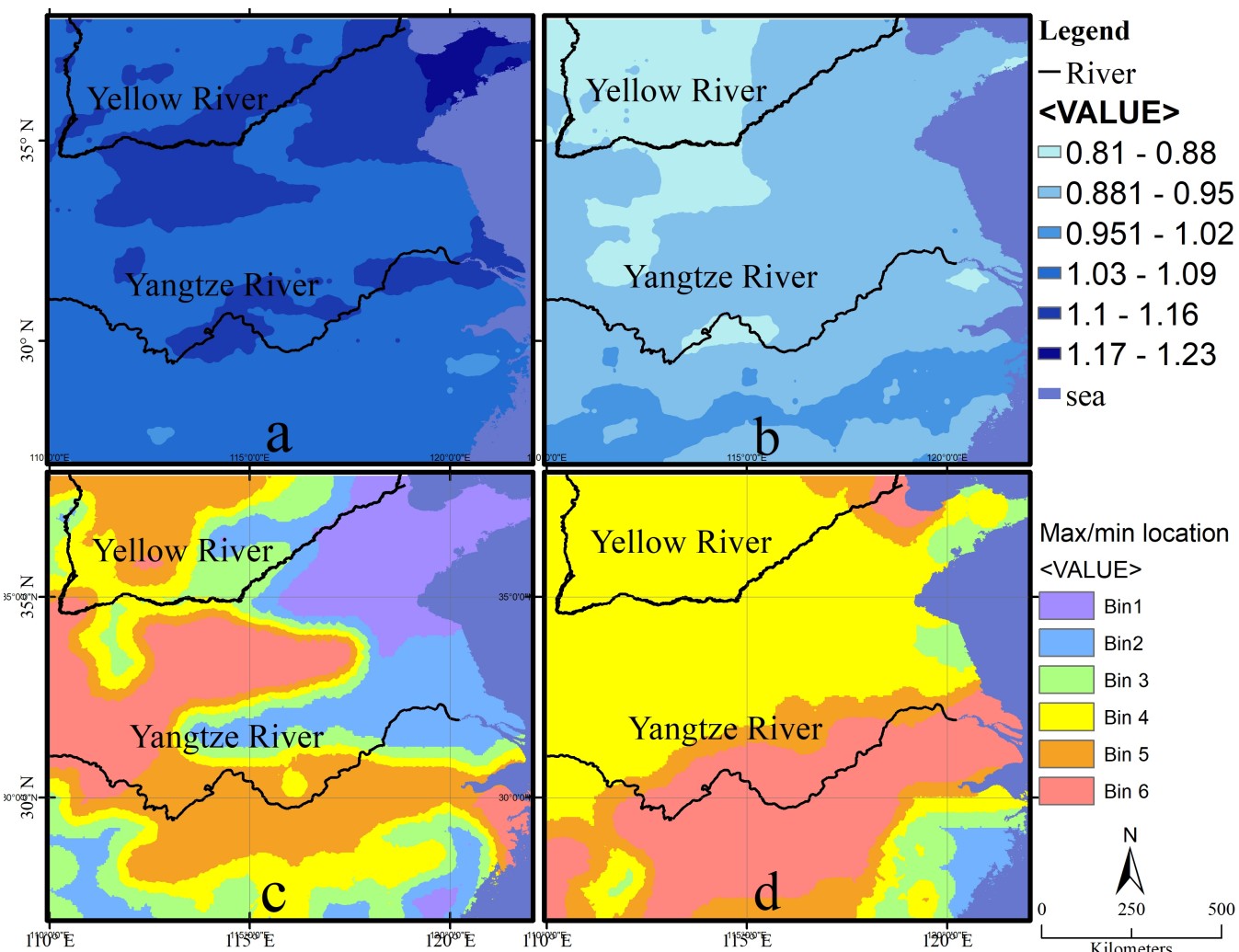

**Figure 8. Geografy of ENSO phase influence.** Maximum and minimum precipitation values in the six phase bins, given by the low-frequency ENSO phase; (a) maximum precipitation value, (b) minimum precipitation value, (c) ENSO phase bin in which maximum precipitation occurs, (d) ENSO phase bin in which minimum precipitation occurs.

ENSO component (cf. Fig. 3c). In YWRB and adjacent areas the minima are located in the positive state of the low-frequency cycle (bins 3, 4, see Fig. 3c). The pattern of localization of the precipitation conditional means maxima in the six phase bins is more complex (Fig. 8c). For better understanding we present precipitation conditional means in all six phase bins for selected gridpoints in Fig. 9.

The results for a representative grid point from the YWRB, markedly influenced by the phases of ENSO oscillatory components, are presented in Fig. 9a. Since each of the six phase bins covering the six-year cycle represents approximately one year in the real time, the conditional means can be considered as an estimate of the amplitude of the precipitation annual cycle (APAC). At the first sight one can see a variation of higher and lower APAC, confirming the well-known alternation of strong

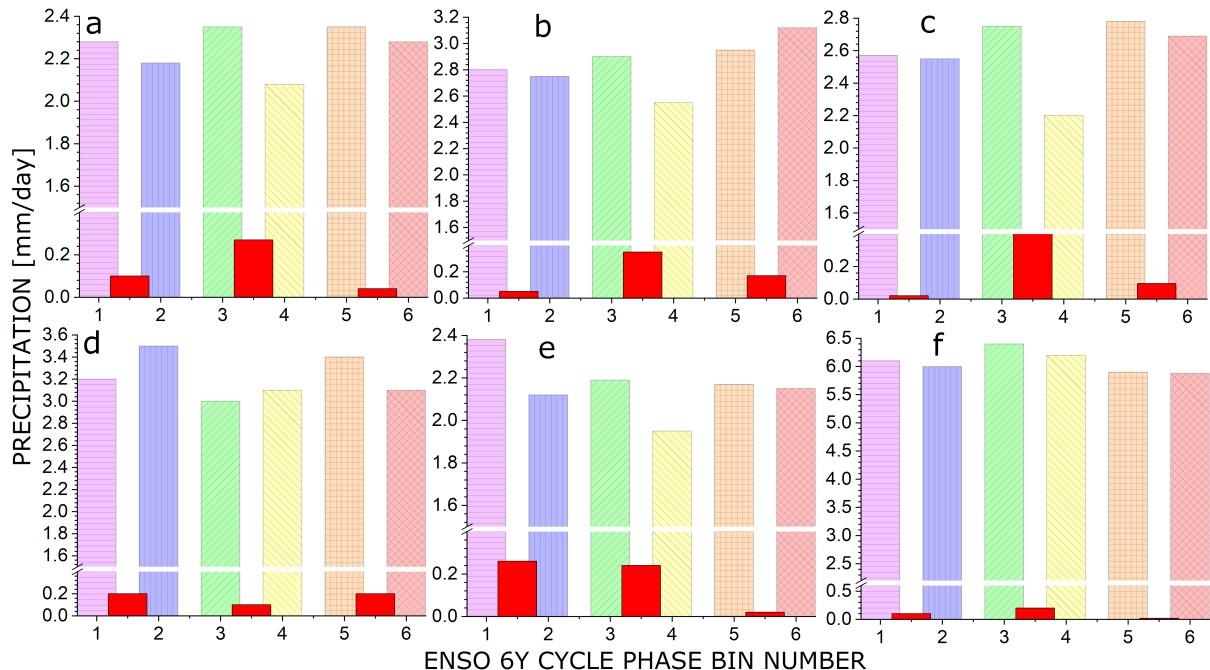

**Figure 9. Local ENSO phase effects.** Precipitation conditional means in the 6 ENSO phase bins at various coordinates: (a) 36°N 110°E, (b) 33°N 114°E, (c) 36°N 111°E , (d) 33°N 120°E, (e) 36°N 117°E, (f) 28°N 117°E. Red color represents difference between two adjacent bins.

and weak monsoon years (Meehl, 1987). The above presented cross-scale causality analysis suggests that this phenomenon is a

consequence of the causal influence of the phase of the ENSO quasi-biennial component on the precipitation annual cycle amplitude. Differences between two adjacent APACs are illustrated by the red bars. The latter can be understood as an estimation of the amplitude of the precipitation quasi-biennial cycle (APQBC). Apparently, APQBC is modulated by the low-frequency (LF, approximately six-year) ENSO component. The maximum APQBC is in the middle, i.e., in the positive state of the LF ENSO cycle (cf. Fig. 3c). This is why both minimum and maximum precipitation conditional means occur in the positive state

of the LF ENSO cycle in this area. Moving south-eastward within the YWRB area, the behaviour of CM is slightly different (Fig. 9b) reminding that the ENSO component driving APAC is not exactly biennial, but quasi-biennial. Its interactions with the precipitation annual cycle lead to frequent phase shifts, disturbing or reversing the weak-strong monsoon year alternation sequence. The minimum CM remains in the positive state of the LF ENSO cycle, however, the maximum CM moves to the last bin number 6, i.e., to the negative state of the LF ENSO cycle. Let us remind that the maximum CM given by the ENSO

state is located in the ENSO negative state in this area. The CM patterns in the next three points (Fig. 9c–e) are similar, with the minimum CM in the positive state of the LF ENSO cycle (middle bins 3, 4), while the maximum CMs are located either in the negative (bins 1, 6) or neutral (bins 2, 5) state of the LF ENSO cycle.

The strength of the influence of the ENSO LF mode on the APQBC determines the location of precipitation CM extrema within the LF cycle. The strongest synchronization of the APQBC with the ENSO LF mode can be seen in Fig. 9a. Due to

markedly largest APQBC in the positive state of the LF cycle, both precipitation minimum and maximum occur there (bins 3, 4). With weaker synchrony the precipitation maxima move to the neutral or negative state of the LF cycle, while the minimum is kept in the positive state.

On the other hand, in the grid point located southward from the Yangtze River, there is practically no modulation of the CM by the ENSO oscillatory components (Fig. 9f). The maximum CM is located in the positive phase, while the minimum CM

in the neutral phase of the LF ENSO cycle. Let us remind that this is the area where the causal effect of the ENSO amplitude dominates and CMs, conditioned on the ENSO states, reach maxima in the ENSO positive and minima in the ENSO neutral state.

## 4 Conclusions

The El Niño Southern Oscillation (ENSO) is an important global climate variability mode influencing precipitation in Yangtze

(YZRB) and Yelow (YWRB) river basins of eastern China. Considerable effort has been expended to analyze and describe this influence, see Tab. 1, and related references (Yang et al., 2005; Li and Zeng; Zhang et al., 2013; Xiao et al., 2015; Zhang et al., 2017; Gao and Wang, 2017; Cao et al., 2017; Chang et al., 2016; Hardiman et al., 2018; Lv et al., 2019; Liu et al., 2020). In such analyses, ENSO, as the cause variable, is characterized using sea surface temperature (SST)-based indices and/or ENSO states, derived from these indices or their anomalies. ENSO is a recurring phenomenon, considered as an irregular

oscillation with variable period and amplitude. Jajcay et al. (2018) decomposed the ENSO dynamics into quasi-periodic modes which mutually interact and temporarily synchronize giving the rise to extreme ENSO events. In this study we analyze possible causal influence of instantaneous phases of ENSO oscillatory components, extracted from the Niño3.4 index using the complex continuous wavelet transform (CCWT), on precipitation variability in different time scales. The precipitation data from eastern China are also decomposed using CCWT. We have detected, with a statistical significance, a causal influence of the phase of

the ENSO quasi-biennial (QB) mode on the amplitude of the precipitation annual cycle. As the second statistically significant causal relation, the amplitude of the precipitation QB mode is influenced by the phase of the ENSO low-frequency (LF) mode with periods between 4 and 7 years, mostly concentrated around the period of 6 years.

The conditional mutual information with the surrogate data testing provide a statistical evidence for the existence of the above described causal relations. In order to estimate their effect in physical quantities, we employ the method of conditional

means (CM) (Jajcay et al., 2016). Precipitation CM were conditioned either on the ENSO states (positive, negative and neutral), thus estimating the causal effect of the ENSO amplitude, or, the effect of phases is assessed by dividing the LF (approximatelly 6 year period) ENSO cycle into six bins equidistantly defined using the phase of the ENSO LF mode. The instantaneous phases of the ENSO oscillatory modes have been found the principal cause variable influencing the precipitation variability in the YWRB area. In YZRB and adjacent areas the ENSO amplitude dominates the causal influence on the precipitation variability.

In the latter area the phase-conditioned CMs do not reflect any influence of the ENSO oscillatory modes and CM maxima and minima occur in positive and neutral, respectively, states of the LF ENSO mode, in agreement with the culmination of the precipitation maxima in ENSO positive states and occurrence of precipitation minima in the ENSO neutral states.

On the other hand, in YWRB the phase-conditioned CMs are modulated by both QB and LF ENSO modes. The cross-scale information flow from the QB mode to the amplitude of the precipitation annual cycle causes the alternation of strong and weak monsoon years (Meehl, 1987). The time scale of this cause is not exactly biennial, but quasi-biennial, therefore its interactions with the precipitation annual cycle lead to frequent phase shifts, disturbing or reversing the weak-strong monsoon year alternation sequence. The differences between the adjacent CM represent the amplitude of the precipitation QB mode which is apparently modulated by the LF (approximately 6 year) ENSO mode. The strength of this modulation determines the position of precipitation CM maxima within the ENSO LF cycle, while the precipitation CM minima occur in the positive state of the ENSO LF cycle. When the precipitation CM are evaluated using the ENSO states, the minima are located in the ENSO neutral and maxima in the ENSO negative state. This inconsistency can be explained by the observation (see Fig. 4 in Jajcay et al. (2018)) that the ENSO extreme events do not necessarily coincide with the minima or maxima of the ENSO LF cycle, but are determined by intermittent synchronization of ENSO QB modes.

Some recent studies (Yu et al., 2022a, b) report a robust contribution of Tropospheric Biennial Oscillations (TBO) to the East Asian summer monsoon transitions. What is the relation between TBO and the QB mode, extracted from the ENSO dynamics, is a challenge for further research. Other studies, e.g., Xiao et al. (2015) observed that, besides ENSO, also North Atlantic Oscillation, Indian Ocean Dipole and Pacific Decadal Oscillation have effect on seasonal precipitation regimes in the Yangtze River Basin. Our cross-scale information flow detection method can be used to establish a causal relationship between precipitation and other large-scale climate variability modes as well, or applied to uncover causal interactions in diverse Earth sciences problems involving multiple time scales. In a broader perspective, the framework we used here is applicable to analyzing phenomena across a wide range of disciplines — for example, in neuroscience, the cross-frequency phase–amplitude coupling has recently been observed in electrophysiological signals reflecting the brain dynamics (Canolty and Knight, 2010) and the conditional mutual information has proven to be a robust tool for its detection (Arinyo-i-Prats et al., 2024).

The fact that complex evolution of climate, atmosphere, or circulation regimes is influenced by interactions of dynamics on multiple time scales is known (Muñoz et al., 2017; Zhang et al., 2023). For instance, Muñoz et al. (2015) suggest that cross-time scale interactions between different climate drivers improves the predictive skill of extreme precipitation. Hsu et al. (2023) show that multiscale interactions, in particular, scale interactions between the monsoon mean field, two modes of intraseasonal oscillation, and synoptic disturbances, were driving the devastating floods in Henan Province, China, during July 2021. Liu et al. (2023b) used the multiscale window transform (MWT) and MWT-based energy and vorticity analysis (MS-EVA), to identify three scales fields: basic-flow fields (>64 days), intraseasonal oscillation fields (8–64 days) and synoptic-scale-eddy fields (<8 days), responsible for the torrential rainfall event, which hit Zhengzhou on July 20, 2021. Ungerovich et al. (2023) emphasize the role of the large scale circulation anomalies associated with ENSO teleconnections in simulation of extreme rainfall events in Uruguay, while Pineda et al. (2023) suggest that the early onset of heavy rainfall on the northern coast of Ecuador in the aftermath of El Niño 2015/2016 was favored by the convective environment in late January due to cross-time-scale interference of the very strong El Niño event and a strong and persistent Madden-Julian oscillation. The presented research, however, is a first step in developing a methodology able to establish a solid statistical evidence for existence of

cross-scale causal interactions and to estimate their effect in measurable, physical quantities. In particular, the results presented here can open a new direction in understanding and predicting precipitation anomalies in eastern Asia. Although physical mechanisms explaining the observed cross-scale information transfers are yet to be established, the uncovered causal relations can already be used in statistical or machine learning tools for forecasting precipitation anomalies. In related considerations, Muñoz et al. (2023) propose to find "Windows of Opportunity" in forecasts across timescales by combining wavelet spectral analysis and a non-stationary time-frequency causality analysis. Materia et al. (2024) try to understand the causal factors behind these windows of opportunity using Liang-Kleeman information flow (Liang, 2013). This study demonstrates the ability to identify sources of cross-scale predictability by using complex continuous wavelet transform and information-theoretic approach to causality (Paluš, 2014).

*Code availability.* FORTRAN codes for CMI estimation are available at http://www.cs.cas.cz/mp/projects/sw/. The scale-wise decomposition was performed using the complex continuous wavelet transform algorithm of Torrence and Compo (1998), codes available at https://github.com/ct6502/wavelets. Simple FORTRAN codes for conditional means or circularly shifted surrogate data are available from the corresponding author M. Paluš on reasonable request.

*Data availability.* The EASMI-ZQY data are available from the corresponding author G. Wang on reasonable request. All other data can be obtained from the web sites listed above in the data description. Some sites require registration.

*Author contributions.* M.P. and G.W. conceptualized and supervised the study, M.P. performed causality analysis and introduced CM computation for selected station data, K.F. computed CM for station data, Y.L. computed CM for the gridded data and created the resulting maps, G.W. and K.F. analyzed the results and M.P. and Y.L. wrote and edited the manuscript. All authors reviewed the manuscript.

*Competing interests.* The authors declare that they have no competing financial interests.

*Acknowledgements.* This study was supported by the Czech Academy of Sciences, Praemium Academiae awarded to M. Paluš, National Key R&D Program of China (2023YFC3007700), the National Natural Science Foundation of China (42075054), and by the Czech–Chinese Academies of Sciences Mobility Plus Project NSFC-23-08.

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
