# Peer review of "Cross-scale causal information flow from El Niño Southern Oscillation to precipitation in eastern China"

_EGUsphere, 2024_

## Referee Comment (RC1)

The authors present a statistical method to relate the NINO3.4 index to precipitation in Eastern China based on wavelet transform. The technique reveals a statistical relation to the phase of the quasi-biennial mode to the variability of the monsoon that suggest that there might be a physical mechanism linking the two. While the general idea of the paper seems clear to me, several details, especially related to figures and discussion are problematic and should be addressed before considering publication.

1) Fig 1. Is rather a table more than a figure, and it would be better if it was considered so. Also, graphically, it makes no sense to have it as a low-resolution figure. Also, the colors do not provide any information. I suggest to re-elaborate it as a simple table. The first reference Yang et al. 2004 has a typo and, in the reference list, it has no year associated. I suggest the authors to revise the other references as well.

2) The term quasi-oscillatory, which is used very often, is unclear. What does that mean? It gives the idea that these components "almost oscillate", however these components are wavelet projections so they oscillate by design. Do the authors mean quasi-periodic? This would be consistent with the use of wavelets, that have a narrow bandwidth in the Fourier space.

3) Data: the authors use ERA 5 reanalysis for the precipitation and analyze it to the single grid cell level. I think the authors should talk about the several limitations of reanalysis products for precipitation and why they believe that these limitations are not an obstacle for their study

4) Figure 4 is problematic because in NINO3.4, the 97-98 Niño should not be chopped. It has another, pointier shape. This should be true both using ERA5 and observations. That timeseries seems to be obtained with raw data rather than anomalies (and therefore should not be addressed as Nino3.4). What I mean is that that the time series seems to be obtained averaging raw SST in the Nino3.4 area, and afterwards normalized to 0 mean (and 1 variance?). It should be explained in the text and, if so, not referred to as NINO3.4 index. Moreover, it is not clear how the Niño and Niña events are defined, if using this timeseries or if they were taken from some other source. If they are taken from this timeseries, then most likely they are not 100% right.

5) It is unclear what the panel d) is representing, the main text reads "*the causal influence of the phase of the 6-year component obtained from the Niño3,4 time series, on the precipitation amplitude for the variability in the quasi-biennial scale*". So the second part of the sentence is unclear, what is the *"precipitation amplitude....quasi-biennal scale"*? Why isn't the precipitation index used again as in the previous case? It would have been more consistent. Also, it is not clear here how the surrogates are computed, are they time-shift surrogates of the Nino3.4 or of the phases timeseries? I think the only reasonable choice is the first (if it does not produce abrupt discontinuities) because the latter for sure produces an abrupt discontinuity so wouldn't be a good surrogate at all for the phases time series.

6) Fig. 5 panels b) and e) are very confusing and misleading. The broken vertical axes hide the fact that the values are indeed very similar. The black and red bars to represent the differences make no sense to me, and perhaps those figures are not needed at all, and a table would be more readable. The label of panels c) and f) reports "histogram" which is rather odd, also because the figure does not look like an histogram. Regarding to this part, the value 0.264 is reported as significant in the text but the p-value is not given. Moreover, it should be probably be corrected (something like a Bonferroni correction) as here the authors are testing multiple differences, implicitly, 5 * 6 / 2 differences.

7) Fig. 8c is either warrying or I didn't understand what it represents. The space is supposed to be divided in a regular grid representing ERA5 grid cells. Then, in each grid cell, rainfall data is analyzed and the ENSO state when the peak precipitation occurs most often marks the cell color. If so, why the blue squares do not fit the grid? How have they been obtained? The situation in Fig. 9 is even worse.

8) Fig. 10. What I said for Fig. 5 panels b) and e) holds here too.

9) *"Although physical mechanisms explaining the observed cross-scale information transfers are yet to be established, the uncovered causal relations can already be used in machine learning tools for forecasting precipitation anomalies."* I think that the first sentence should be expanded. The authors should provide an idea for further investigations of what could be a possible mechanism that makes the low frequency of ENSO impact the monsoon at the 2ys scale, referring to the literature that surely has already investigated the link between ENSO and China monsoon from another angle. The second sentence is very vague and does not provide any valuable information.

10) If the QB mode is the one that impacts the dataset the most why, to make all figures up to Fig 5, it was used the 6ys mode instead?

Minor:

The sentence *"other recent follow-up studies were particularly concerned with the predictability and future of ENSO projections attributable to global to regional scale interconnections, including the combined influence of ENSO and PDO"* is unclear.

*"The Yangtze River is China's longest river and the world's third largest, contributes considerably to China's equitable economic and ecological growth"*, I understand economic growth but what does ecological growth mean?

*"The YZRB is predominantly controlled by Siberian northwest winter and southeast summer monsoon."* Written like this it suggests the existence of a Siberian monsoon. Perhaps the authors are mentioning the Siberian High?

Fig. 2 carries almost no information. In the text, a lot of geographical details are given *"hydrological station (YHS) separates the Yangtze River into upper and lower sections and is renowned as the 'Gateway to the Three Gorges'. The Three Gorges Dam (TGD) lies just approximately 40 kilometers above (Xu et al., 2007). The territory above Yichang station is commonly regarded as the upper sub-basin of the YZRB; the region from Yichang station and Hukou station is the middle sub-basin; and the region under Hukou station is the lower sub-basin of the YZRB (Fang et al., 2018). The YZRB lies in subtropical and temperate climate zone dominated by monsoonal winds; the southern region exhibits subtropical climate while northern region presents temperate zone. Major flooding in YZRB is linked with warm ENSO and strong summer monsoons typically occur after El Niño conditions in the winter, while weak winter"* which are impossible to locate in the maps, making both the map and this paragraph useless.

Fig 3. To use red for precipitation is a semiotically unfortunate choice. Moreover, one cannot see the rivers.

Eq.1 Exp parenthesis is too small

The sentence *"The causal delay can be found in the causality analysis"* in that part of the text is particularly unclear.

Nino3.4 sometimes is written with ~ (often in the text) and sometimes not (e.g. label of fig.4). The authors should be consistent: either with or without. Sometimes, it is written with a coma ("Nino3,4") the point should be employed instead.

In Fig5. The panels would benefit of some labels.

Fig. 6 Time unit of measure should be years, plural, not year. The colorbar needs a label

Fig 7. "a)", "b)", "c)" and "d)" are barely visible as the rivers and their names. The crosses hide the information about the difference patterns because they are too heavy.

Fig. 8. And 9. Using the same color scale for maximum and minimum precipitation makes very hard to see any differences between the two maps (especially in 8), aside the mean level. Again, rivers colors are unfortunate. "c)" and "d)" labels are barely visible. In fig 9 the color scale of the legend of c and d should be periodic, as the numbers from 1 to 6 represent phases of the 6ys mode, so close numbers should have "close" colors and 1 and 6 should be regarded as close among them.

Fig. 10 Labels are oversized compared to panels, a line break or a smaller character dimension should be used.

In the sentence *"for example, in neuroscience, where cross-frequency phase–amplitude coupling has recently been observed in electrophysiological signals reflecting the brain dynamics"*, I think that mentioning neurosciences here is really out of context given the journal.

---

## Author Comment (AC1)

**Response to Review # 2 for the manuscript:**
**Cross-scale causal information flow from El Niño Southern Oscillation to precipitation in eastern China**

Yasir Latif[1], Kaiyu Fan[2,3], Geli Wang[2,*], and Milan Paluš[1,*]

[1]Department of Complex Systems, Institute of Computer Science of the Czech Academy of Sciences, 182 00 Prague 8, Czech Republic
[2]Key Laboratory for Middle Atmosphere and Global Environment Observation, Institute of Atmospheric Physics, Chinese Academy of Sciences, Beijing 100029, China
[3]Dalian Meteorological Bureau, Dalian 116001, China
[*]Joint corresponding authors

**Correspondence:** Geli Wang (wgl@mail.iap.ac.cn) and Milan Paluš (mp@cs.cas.cz)

**Dear Editors & Reviewers,**

We would like to thank the reviewers for careful reading and insightful comments which give us the opportunity to improve the manuscript. There were many suggestions to change the figures, including the scale for the precipitation maps. Therefore we present relevant (sub)figures in two different scales, blue and red-yellow (the original) and ask the reviewers and editors to advice which one to use. In order to avoid confusion, we use the same numbers of figures (2–10) as in the original submission. In a future revised manuscript the numbers of figures will be decreased by one, since the former Fig. 1 will be presented as Table 1, as requested by a reviewer.

In the following there are our answers to the comments of the Reviewer # 2. Reviewer's comments are in *blue italics*, while our answers in black roman font. If we cite a text from the manuscript, the changes are in red color.

RC2: 'Comment on egusphere-2024-400', Anonymous Referee #2, 26 Mar 2024

Review of the manuscript "Cross-scale causal information flow from El Niño Southern Oscillation to precipitation in eastern China" by Yasir Latif , Kaiyu Fan, Geli Wang and Milan Paluš.

*Authors present original research on the link between ENSO states and/or quasi-oscillatory (QO) ENSO components and QO Chinese-river-basins precipitation components obtained by a wavelet approach. Authors use an information-theoretic approach to establish causal links and optimal delays between ENSO and precipitation. They also estimate statistical significance causality thresholds using means over surrogates preserving the spectrum. The method and results are interesting and susceptible of development and generalization by using other possible atmospheric-oceanic indexes as drivers of precipitation. The material of the manuscript is acceptable for publication after the adjustment of several points that can improve much better the presented research.*

We would like to thank the reviewer for the careful reading of the manuscript, its positive evaluation and valuable advices to improve the manuscript.

*Major points:*

*Fig. 2 A color bar for the orography height must be included. Add in the maps the location of local precipitation stations used in manuscript.*

Fig. 2 was modified as suggested, see below.

[Figure]

**Figure 2. Study area.** (Top) Localization of the selected region in Yangtze and Yellow River Basins. (Bottom) Study area in a detailed view, including the positions of selected stations.

*Fig. 4a. The ENSO time-series (or a proxy of that) is plotted (black curve) with indication of positive, negative, and neutral ENSO states. However, it is the ONI (lines 117-120) that is used to discriminate ENSO states. From the graph the cross of +0.5 and -0.5 to determine ENSO phases is not well suited. Change the graph accordingly with the chosen criterium of ENSO phases.*

The data presented in the original form of Fig. 4 was normalized "raw", i.e., not anomalized data marked as Nino3.4 at https://www.cpc.ncep.noaa.gov/data/indices/ersst5.nino.mth.91-20.ascii. For better understanding of the definition of the ENSO states, in the new version of the Figure 4a we present anomalized Nino3.4 data as well as the ONI index used for the states definition. We present the new version of Fig. 4 below.

[Figure]

**Figure 4. ENSO states and binning of the low-frequency cycle.** From top to bottom: (a) A segment of anomalized Niño3.4 (black) and ONI (gray) time series with marked ENSO states: warm episodes ENSO+ (light red), cold episodes ENSO- (light blue) and neutral ENSO0 state (white). (b) The same segment of anomalized Niño3.4 time series (black) with its CCWT-extracted 6-yr component (blue) and the instantaneous phase (red) of the latter. (c) The 6-yr Niño3.4 component (blue) and its instantaneous phase (red). The bars of different colors and patterns mark the 6 phase bins into which each 6-yr cycle is divided. (d) A segment of reanalysis precipitation data from the gridpoint 33.75°N 115.75°E (black) and the 4 months lagged phase (red) of the 6-yr Nino3.4 cycle and related phase bins (bars of different colors and patterns) in which the precipitation conditional means are computed.

*Line 171. The formula of the transfer-entropy (TE) (eq. 9) is proposed by Wibral et al. (2013) as a CMI in the case of SPO (Self Prediction Optimality) of Y states prior to the forecast delay tau. This is a very conservative estimate of TE since the SPO may be never reached with TE of eq.9 being underestimated. Authors shall comment on this.*

Thank you very much for pointing to this issue. We can add an explanation why the Wibral formula is used for the causal delay estimation, while the original formulation 7, 8 for the statistical testing of the presence of causality:

The Wibral et al. (2013) formula 9 is used in order to establish the causal delay, while the formulas 7 and 8 are used for testing the statistical significance of uncovered causal relations. Wibral et al. (2013) formula 9 was proposed as a CMI in the case of Self Prediction Optimality (SPO) of $y$ states prior to the forecast delay $\tau$. This is a very conservative estimate of CMI/TE since the SPO may be never reached with CMI/TE of eq. 9 being underestimated. CMI estimated according to eq. 7 or 8 is more sensitive with respect to detection of causality.

*Line 188. The Z statistics use Id and Is. Id is the CMI value estimated from the studied data, Is is the mean for 100 real-izations of the surrogate data. What formula is used for Id and Is? Are formulas 8 or 9 used? Give an example used in the manuscript.*

As stated above, formula 7/8 is used for testing, i.e. the Z-scores in Fig. 6 were obtained using eqs. 7 or 8. Eq. 9 was used only for the estimation of causal delay in Figs. 5a,d. More details will appear in the revised manuscript as:

Computing the conditional means, the precipitation time series is not exactly aligned in time with the ENSO states or the ENSO phase bins, since the causal effect of ENSO can occur with some time delay. The causal delay can be found in the causality analysis as follows: In Fig. 5a the conditional mutual information represents the causal influence of ENSO states on the precipitation (EASMI-ZQY index). It was computed using the Wibral et al. (2013) formula 9 in which $x(t)$ is a discrete 3-valued function of the ENSO states, $y(t)$ is the precipitation EASMI-ZQY index discretized into four bins using the equi-quantal binning algorithm (Paluš and Vejmelka, 2007), $d_2 = 1$.
.....
Fig. 5d shows the conditional mutual information showing the causal influence of the phase of the 6-year component obtained from the Niño3,4 time series, on the precipitation amplitude for the variability in the quasi-biennial scale (blue). It was again computed using the Wibral et al. (2013) formula 9, however, since the cross-scale causality is evaluated, before applying Eq. 9, the Niño3.4 and precipitation data underwent CCWT and now $x(t)$ is the ENSO phase $\phi_f(t)$ for the frequency $f$ related to the period 6 years and $y(t)$ is the precipitation amplitude $A_f(t)$ for the frequency $f$ related to the period 2 years. $d_2 = 3$ and the Gaussian estimator (Paluš, 2014) is used.

*Line 202-205. Conditional means of precipitation, given the phase of ENSO CCWT component (or the ENSO state) are computed on a grid basis. However on Figs. 8c,d and Fig 9c,d the values are overlapped over the grid cells. Figures must be redone precisely.*

The grids which appeared in the first versions od figures were used to avoid interpolating of the mapped values. While mapping precipitation values, the interpolation has a physical sense, in mapping extrema occurrence in bins 1–6 we need to have only integer values, while interpolation does not have any sense. In the new version of figures we made the maps in a different way and the misleading grids did not appear. See the new versions of figures bellow.

*Line 220. What authors conclude from the histogram of Fig.5c?*

It was stated in the lines 220–221 of the original manuscript:

"It can be observed that the red bar lies inside the surrogate histogram which means the difference between two ENSO states is not statistically significant in this grid point."

In other words, the difference of this value can occur by chance, as the surrogate results suggest.

*Fig. 5 Authors compute: The conditional mutual information measuring the causal influence of ENSO states on precipitation characterized by the EASMI-ZQY index (solid blue line) and causality in the opposite direction (dashed black line). What exact formulas authors use? (Eqs 8,9?). What are the embedding dimensions used? Be precise about X and Y in this case. By opposite direction, what authors mean? X, Y are swapped? Give a mathematical expression in the method section. Authors use certain precipitation stations. What was the criterium of choice? Explain. The values of significance values (red thresholds9 on panels a) and d) are not very well explained how they are computed. Explain it in detail in the method section.*

The formulas and embedding dimensions are specified as follows:

The causal delay can be found in the causality analysis as follows: In Fig. 5a the conditional mutual information represents the causal influence of ENSO states on the precipitation (EASMI-ZQY index). It was computed using the Wibral et al. (2013) formula 9 in which $x(t)$ is a discrete 3-valued function of the ENSO states, $y(t)$ is the precipitation EASMI-ZQY index discretized into four bins using the equiquantal binning algorithm (Paluš and Vejmelka, 2007), $d_2 = 1$.
.....
Fig. 5d shows the conditional mutual information showing the causal influence of the phase of the 6-year component obtained from the Niño3,4 time series, on the precipitation amplitude for the variability in the quasi-biennial scale (blue). It was again computed using the Wibral et al. (2013) formula 9, however, since the cross-scale causality is evaluated, before applying Eq.

9, the Niño3.4 and precipitation data underwent CCWT and now $x(t)$ is the ENSO phase $\phi_f(t)$ for the frequency $f$ related to the period 6 years and $y(t)$ is the precipitation amplitude $A_f(t)$ for the frequency $f$ related to the period 2 years. $d_2 = 3$ and the Gaussian estimator (Paluš, 2014) is used.

The opposite direction indeed means swapped variables, as was explained in equations 7 and 8.

Figure 5 is a part of the Data and Methods section, the particular gridpoint was chosen as an example. The full grid of data is reported in Figure 7.

For the signidicance criterion we will add to the Fig. 5 caption:

The red line is the significance threshold given as the mean+2SD for the surrogate data.

*Fig. 5 In this figure the authors choose a point in the North where the ENSO states does not discriminate significantly the precipitation (see crosses in Figs. 7b,d). However in the South region (Yangtze River basin) it is apparent the existence of locations where ENSO states have an important role on precipitation. Authors should add the equivalent of Figs. 5a,b,c for a particular significant location in the South.*

Figure 5 is a part of the Data and Methods section, the particular gridpoint was chosen as an example. The full grid of data is reported in Figure 7.

*Figure 6 presents the Z score of a certain CMI (eq. 7) between the instantaneous amplitudes of the CCWT of El-Niño and of the precipitation. Which is the value of lag tau used? Clarify. Values presented in Fig. 6 depend on tau. Explain.*

The end of the subsection **2.6 Conditional mutual information as a causality measure** will be completed as follows:

... while the formulas 7 and 8 are used for testing the statistical significance of uncovered causal relations. For testing the cross-scale causality, before applying Eq. 8, the Niño3.4 and precipitation data underwent CCWT and now $x(t)$ is the ENSO phase $\phi_{f_i}(t)$ for a particular frequency $f_i$, and $y(t)$ is the precipitation amplitude $A_{f_j}(t)$ for a frequency $f_j$. The Gaussian estimator was used and $d_2 = 3$ was chosen as in (Paluš, 2014) based on "saturation of the results", i.e., obtaining unchanged results for $d_2 = 4$ in comparison with $d_2 = 3$. The tested value is the CMI average for time lags $\tau = 1$ to 6 months, according to the recommendation in (Paluš and Vejmelka, 2007).

*Fig. 6 uses values for 6 stations but the geographical coordinates are not given. Provide them in a Table in the data section and point them in the first maps.*

The table will be added, see also the new version of Fig. 2. The six averaged stations include also two individually presented stations.

**Table 1.** Geographical coordinates of seven local precipitation stations used in the combined region of Yellow and Yangtze River basins

| Station ID | Station name | Province | Longitude | Latitude |
| --- | --- | --- | --- | --- |
| 57355 | Huangjiawan | Hubei | 110.4 E | 31 N |
| 57494 | Wuhan | Hubei | 114.1 E | 30.6 N |
| 57598 | Hejiadian | Jianxi | 114.6 E | 29 N |
| 57799 | Yankeng | Jiangxi | 114.9 E | 27.1 N |
| 58102 | Bozhou | Anhui | 115.7 E | 33.7 N |
| 58457 | Hangzhou | Zhejiang | 120 E | 30.2 N |
| 58527 | Jingdezhen | Jiangxi | 117.2 E | 29.3 N |

*Figure 7. Authors present the effects of two causal mechanisms. Concerning the effect of oscillatory components of ENSO, only results for the 6-year component are presented in a map. Since the QB component is also relevant in the annual cycle of precipitation, authors shall conceive a map (similar to Fig. 7a,c) giving the representativeness of such link.*

In causality analysis, we report the effects of both QB and LF modes. In the conditional means analysis, using the six bins of the LF phase, both effects are apparent, as described in the paper: The effect of the QB mode results in alternating strong and weak years (bins), while the LF affects the precipitation-QB amplitude, i.e. the difference between the adjacent bins of the 6yr phase. Focusing on the QB mode alone, one would effectively average all even and odd bins and the LF mode would attenuate the effect of the QB mode. The present methodology estimates the combined effect of both modes. In principle we could distinguish, in each grid point, which mode has a stronger effect. We are afraid, however, that the manuscript is already very complex to understand, so that we leave such distinction for further research.

*Conclusion: Authors say: 'physical mechanisms explaining the observed cross-scale information transfers are yet to be established, the uncovered causal relations'. Can authors elaborate a little bit more here,*

We will extend the concluding paragraph as follows:

The fact that complex evolution of climate, atmosphere, or circulation regimes is influenced by interactions of dynamics on multiple time scales is known (Muñoz et al., 2017; Zhang et al., 2023). For instance, Muñoz et al. (2015) suggest that cross-time scale interactions between different climate drivers improves the predictive skill of extreme precipitation. Hsu et al. (2023) show that multiscale interactions, in particular, scale interactions between the monsoon mean field, two modes of intraseasonal oscillation, and synoptic disturbances, were driving the devastating floods in Henan Province, China during July 2021. Liu et al. (2023) used the multiscale window transform (MWT) and MWT-based energy and vorticity analysis (MS-EVA), to identify

three scales fields: basic-flow fields (>64 days), intraseasonal oscillation fields (8–64 days) and synoptic-scale-eddy fields (<8 days), responsible for the torrential rainfall event, which hit Zhengzhou on July 20, 2021. Ungerovich et al. (2023) emphasize the role of the large scale circulation anomalies associated with ENSO teleconnections in simulation of extreme rainfall events in Uruguay, while Pineda et al. (2023) suggest that the early onset of heavy rainfall on the northern coast of Ecuador in the aftermath of El Niño 2015/2016 was favored by the convective environment in late January due to cross-time-scale interference of the very strong El Niño event and a strong and persistent Madden-Julian oscillation. The presented research, however, is a first step in developing a methodology able to establish a solid statistical evidence for existence of cross-scale causal interactions and to estimate their effect in measurable, physical quantities. In particular, the results presented here can open a new direction in understanding and predicting precipitation anomalies in eastern Asia. Although physical mechanisms explaining the observed cross-scale information transfers are yet to be established, the uncovered causal relations can already be used in statistical or machine learning tools for forecasting precipitation anomalies. In related considerations, Muñoz et al. (2023) propose to find "Windows of Opportunity" in forecasts across timescales by combining wavelet spectral analysis and a non-stationary time-frequency causality analysis. Materia et al. (2024) try to understand the causal factors behind these windows of opportunity using Liang-Kleeman information flow (Liang, 2013). This study demonstrates the ability to identify sources of cross-scale predictability by using complex continuous wavelet transform and information-theoretic approach to causality (Paluš, 2014).

*Minor points:*

*Line 53. After the work of Jajcay et al. (2018), it is worth citing the work of Pires et al. (2021) which also studies the interactions between quasi-oscillatory ENSO components and the role of triadic resonances and synchronization in the explanation of super El-Niños.*

*Pires C.A. and Hannachi A. (2021) Bispectral analysis of nonlinear interaction, predictability and stochastic modelling with application to ENSO, Tellus A: Dynamic Meteorology and Oceanography, 73:1, 1-30, DOI: 10.1080/16000870.2020.1866393*

Thank you very much for bringing this reference which will be added in the revised manuscript.

*Eq. (1) time t is missing in the complex exponential.*

Thank you very much for pointing this omission, we will correct as follows:

$$\psi(t) = \frac{1}{\sqrt{2\pi\sigma_t^2}} \exp\left(-\frac{t^2}{2\sigma_t^2}\right) \exp\left(2\pi i f t\right), \qquad 1$$

There is $\sigma_f = 1/\pi \sigma_t$.

*The use of red color for the rivers is not ideal since the color bar includes red in several figures. Suggestion: use black.*

All the maps were modified and two version in two different color scales are given for assessment of the referees which is more appropriate.

[Figure]

**Figure 3. Precipitation and its variability in the study area.** Spatial distribution of precipitation and its variability during 1951-2020; (a) mean precipitation, (b) precipitation standard deviation (SD), (c) relative difference between ENSO positive and ENSO neutral state, (d) relative precipitation SD (SD/mean precipitation).

We would like to thank again the reviewer for the suggestions which helped us to improve the manuscript.

[revised manuscript text omitted]

---

## Author Comment (AC2)

**Response to Review # 1 for the manuscript:**
**Cross-scale causal information flow from El Niño Southern Oscillation to precipitation in eastern China**

Yasir Latif[1], Kaiyu Fan[2,3], Geli Wang[2,*], and Milan Paluš[1,*]

[1]Department of Complex Systems, Institute of Computer Science of the Czech Academy of Sciences, 182 00 Prague 8, Czech Republic
[2]Key Laboratory for Middle Atmosphere and Global Environment Observation, Institute of Atmospheric Physics, Chinese Academy of Sciences, Beijing 100029, China
[3]Dalian Meteorological Bureau, Dalian 116001, China
[*]Joint corresponding authors

**Correspondence:** Geli Wang (wgl@mail.iap.ac.cn) and Milan Paluš (mp@cs.cas.cz)

**Dear Editors & Reviewers,**

We would like to thank the reviewers for careful reading and insightful comments which give us the opportunity to improve the manuscript. There were many suggestions to change the figures, including the scale for the precipitation maps. Therefore we present relevant (sub)figures in two different scales, blue and red-yellow (the original) and ask the reviewers and editors to advice which one to use. In order to avoid confusion, we use the same numbers of figures (2–10) as in the original submission. In a future revised manuscript the numbers of figures will be decreased by one, since the former Fig. 1 will be presented as Table 1, as requested by a reviewer.

In the following there are our answers to the comments of the Reviewer # 1. Reviewer's comments are in *blue italics*, while our answers in black roman font. If we cite a text from the manuscript, the changes are in red color.

*The authors present a statistical method to relate the NINO3.4 index to precipitation in Eastern China based on wavelet transform. The technique reveals a statistical relation to the phase of the quasi-biennial mode to the variability of the monsoon that suggest that there might be a physical mechanism linking the two. While the general idea of the paper seems clear to me, several details, especially related to figures and discussion are problematic and should be addressed before considering publication.*

We would like to thank the reviewer very much for the valuable comments. Our responses follow.

*1) Fig 1. Is rather a table more than a figure, and it would be better if it was considered so. Also, graphically, it makes no sense to have it as a low-resolution figure. Also, the colors do not provide any information. I suggest to re-elaborate it as a simple table. The first reference Yang et al. 2004 has a typo and, in the reference list, it has no year associated. I suggest the*

*authors to revise the other references as well.*

We have checked the references and created a table instead of Fig. 1, see below.

**Table 1. Short literature review**. Previous studies regarding the impact of ENSO on annual and seasonal precipitation in Chinese regions. * Et: Evapotranspiration, Ppt: Precipitation, YRB: Yellow River Basin, YZRB: Yangtze River Basin, ERB: East River Basin

| Authors | Region/study period | Variable | Observation |
|---|---|---|---|
| Yang et al., 2004 | YRB* (1951-2000) | Ppt* and Et* | Decreased Ppt and increased Et during 1990-2000 |
| Li and Zheng., 2013 | YRB (1951-2012) | Ppt and ENSO | Decadal weakening of autumn Ppt due to ENSO |
| Zhang et al., 2013 | ERB* (1956-2005) | Ppt and ENSO | Strong correlation between ENSO and April Ppt |
| Xiao et al., 2015 | YZRB* (1960-2019) | Ppt and ENSO | Strong relationship between ENSO and seasonal Ppt |
| Zhang et al., 2016 | YZRB (1979-2015) | Ppt and ENSO | Dominant/predictable impact of ENSO on Asian Ppt |
| Gao and Wang., 2017 | YRB (1960-2011) | Extreme Ppt | Weakening of summer monsoon |
| Cao et al., 2017 | YZRB (1960-2015) | Ppt and ENSO | Strong ENSO impact on wetting and drying Ppt pattern |
| Chang et al., 2017 | YRB (1956-2010) | Ppt and runoff | Abrupt change in Ppt with insignificant trends at 8 stations |
| Hardiman et al., 2018 | YZRB (1992-2015) | Ppt and ENSO | Linear impact of ENSO on summer Ppt |
| Lv et al., 2019 | China (1960-2013) | Ppt and ENSO | Decreased Ppt but increased extreme events attributed to ENSO |
| Liu et al., 2020 | YRB (1961-2017) | Seasonal Ppt | Linear impact of ENSO on winter and spring Ppt |

*2) The term quasi-oscillatory, which is used very often, is unclear. What does that mean? It gives the idea that these components "almost oscillate", however these components are wavelet projections so they oscillate by design. Do the authors mean quasiperiodic? This would be consistent with the use of wavelets, that have a narrow bandwidth in the Fourier space.*

The term "quasi-oscillatory" is typically used for the ENSO phenomenon as a whole, the reviewer is correct that for the wavelet components the term "quasi-periodic" would be more appropriate. We will reformulate the text accordingly.

*3) Data: the authors use ERA 5 reanalysis for the precipitation and analyze it to the single grid cell level. I think the authors should talk about the several limitations of reanalysis products for precipitation and why they believe that these limitations are not an obstacle for their study*

We will add a text:

Recently, some studies focused on comparing the performance of model-based precipitations such as ERA to satellite products for mainland Chinese regions and Tibetan Plateau, since reliable precipitation retrievals with fine spatiotemporal resolutions are vital in global and regional evaluations (Xu et al., 2022; Hu and Yuan, 2021). Model-based precipitation estimates, which are an essential alternative to satellite-based precipitation products, have grown rapidly in recent decades. Model-based products outperform satellite products in subregions of temperate monsoon climate (TM) and temperate continental climate

(TC) (Xu et al., 2022). However, when compared to gauge precipitation, ERA 5 performance was being compromised in terms of frequency and intensity for Tibetan Plateau (Hu and Yuan, 2021). The latter study further argued that rainfall gauges on the Tibetan Plateau are generally positioned in valleys and may not correctly reflect the region's average. Another study for the same regions of TP and Sichuan province observed that ERA-Interim exhibits better performance than IMERG_E, IMERG_L, IMERG_F, CHIRPS, TRMM_3B42, TRMM_3B42RT (Lei et al., 2021). Future study will require additional observations and clarification of station locations and higher levels (Hu and Yuan, 2021). ERA 5 has replaced ERA- Interim and this release offers several improvements over the previous ERA-Interim reanalysis solution due to improved design and generation methodologies. In comparison to ERA-Interim, this dataset is more advanced due to several factors including a high resolution, day-by-day archiving, diverse data sources, better assimilation, and diversified data products (Tarek et al., 2020). The assessment at the monthly flood season (Lavers et al., 2022) indicates that the ERA5 is slightly better than the other models. It is better in the extratropics. ERA5 precipitation has been found to be a sufficiently excessive source of information in the non-tropical areas. Therefore, it is suggested that ERA5 be utilized primarily for extratropical precipitation monitoring. ERA5 performs spatially across China, with the highest correlation coefficient values in eastern, northwestern, and north China and the lowest biases in southeast China (our study area) (Jiao et al., 2021). Similarly, intensity comparisons show strong agreement between ERA-5 and EOBS in Germany, Ireland, Sweden, and Finland, but some disagreement in places with scarce input stations (Rivoire et al., 2021).

*4) Figure 4 is problematic because in NINO3.4, the 97-98 Niño should not be chopped. It has another, pointier shape. This should be true both using ERA5 and observations. That timeseries seems to be obtained with raw data rather than anomalies (and therefore should not be addressed as Nino3.4). What I mean is that that the time series seems to be obtained averaging raw SST in the Nino3.4 area, and afterwards normalized to 0 mean (and 1 variance?). It should be explained in the text and, if so, not referred to as NINO3.4 index. Moreover, it is not clear how the Niño and Niña events are defined, if using this timeseries or if they were taken from some other source. If they are taken from this timeseries, then most likely they are not 100right.*

The used ENSO data are described, including URL addresses from where they were downloaded, in Sec. **2.4 ENSO data**. This section also defines ENSO states ENSO+ (El Niño), ENSO- (La Niña) and ENSO0 (neutral). The referee is correct, however, that the data presented in the original form of Fig. 4 was normalized "raw", i.e., not anomalized data marked as Nino3.4 at https://www.cpc.ncep.noaa.gov/data/indices/ersst5.nino.mth.91-20.ascii. For better understanding of the definition of the ENSO states, in the new version of the Figure 4a we present anomalized Nino3.4 data as well as the ONI index used for the states definition. We present the new version of Fig. 4 below.

*5) It is unclear what the panel d) is representing, the main text reads "the causal influence of the phase of the 6-year component obtained from the Niño3,4 time series, on the precipitation amplitude for the variability in the quasi-biennial scale". So the second part of the sentence is unclear, what is the "precipitation amplitude....quasibiennal scale"? Why isn't the precipitation index used again as in the previous case? It would have been more consistent. Also, it is not clear here how the*

[Figure]

**Figure 4. ENSO states and binning of the low-frequency cycle.** From top to bottom: (a) A segment of anomalized Niño3.4 (black) and ONI (gray) time series with marked ENSO states: warm episodes ENSO+ (light red), cold episodes ENSO- (light blue) and neutral ENSO0 state (white). (b) The same segment of anomalized Niño3.4 time series (black) with its CCWT-extracted 6-yr component (blue) and the instantaneous phase (red) of the latter. (c) The 6-yr Niño3.4 component (blue) and its instantaneous phase (red). The bars of different colors and patterns mark the 6 phase bins into which each 6-yr cycle is divided. (d) A segment of reanalysis precipitation data from the gridpoint 33.75°N 115.75°E (black) and the 4 months lagged phase (red) of the 6-yr Nino3.4 cycle and related phase bins (bars of different colors and patterns) in which the precipitation conditional means are computed.

*surrogates are computed, are they time-shift surrogates of the Nino3.4 or of the phases timeseries? I think the only reasonable choice is the first (if it does not produce abrupt discontinuities) because the latter for sure produces an abrupt discontinuity so wouldn't be a good surrogate at all for the phases time series.*

For better understanding, we extend the text as folows:

Computing the conditional means, the precipitation time series is not exactly aligned in time with the ENSO states or the ENSO phase bins, since the causal effect of ENSO can occur with some time delay. The causal delay can be found in the causality analysis as follows: In Fig. 5a the conditional mutual information represents the causal influence of ENSO states on

the precipitation (EASMI-ZQY index). It was computed using the Wibral et al. (2013) formula 9 in which $x(t)$ is a discrete 3-valued function of the ENSO states, $y(t)$ is the precipitation EASMI-ZQY index discretized into four bins using the equi-quantal binning algorithm (Paluš and Vejmelka, 2007), $d_2 = 1$.

.....

Fig. 5d shows the conditional mutual information showing the causal influence of the phase of the 6-year component obtained from the Niño3,4 time series, on the precipitation amplitude for the variability in the quasi-biennial scale (blue). It was again computed using the Wibral et al. (2013) formula 9, however, since the cross-scale causality is evaluated, before applying Eq. 9, the Niño3.4 and precipitation data underwent CCWT and now $x(t)$ is the ENSO phase $\phi_f(t)$ for the frequency $f$ related to the period 6 years and $y(t)$ is the precipitation amplitude $A_f(t)$ for the frequency $f$ related to the period 2 years. $d_2 = 3$ and the Gaussian estimator (Paluš, 2014) is used. It is evident again that there is no significant causality from the precipitation to the ENSO phase shown by the black dashed line. However, the influence of the ENSO phase on the amplitude of precipitation exhibits a clear significant peak in approximately four months (lags 2-6 months). Therefore, for the calculation of conditional means for the six phase bins, we used the ENSO phase bins having the time shift of four months back relative to precipitation data.

Considering the use of the surrogate data, the sec. **2.7 Surrogate data for statistical testing** will be extended by the sentence:

In the cross-scale analyses, the surrogate data are applied directly to the raw data before the application of the wavelet transform.

*6) Fig. 5 panels b) and e) are very confusing and misleading. The broken vertical axes hide the fact that the values are indeed very similar. The black and red bars to represent the differences make no sense to me, and perhaps those figures are not needed at all, and a table would be more readable. The label of panels c) and f) reports "histogram" which is rather odd, also because the figure does not look like an histogram. Regarding to this part, the value 0.264 is reported as significant in the text but the p-value is not given. Moreover, it should be probably be corrected (something like a Bonferroni correction) as here the authors are testing multiple differences, implicitly, 5 \* 6 / 2 differences.*

The panels 5b and 5e are used to explain the evaluation of the differences between the conditional means. We compute differences between all conditional means in order to find their maximum, we plot, for illustration, differences of the adjacent bins. In the case of 5e, the differences of the adjacent bins have also another meaning – they represent the amplitude of the quasibiennial cycle in the precipitation which is modulated by the ENSO low-frequency mode. This phenomenon is more discussed in Fig. 10.

Panels c) and f) indeed include histograms as estimators of surrogate data distributions. In f, the data value 0.264 lies almost outside the distribution of the surrogate data (the null hypothesis), it is the essence of statistical significance. In fact, it touches the most right bin whose value is 0.006, thus this is the estimate of p-value. Using a finer histogram we could get 0.003 or less, the value is not so important as the position relative to the distribution of the null hypothesis. We test only the maximum

difference, thus here it is a single test. (Maximum is taken from the data, and maxima are taken from all surrogate realizations.)

*7) Fig. 8c is either warrying or I didn't understand what it represents. The space is supposed to be divided in a regular grid representing ERA5 grid cells. Then, in each grid cell, rainfall data is analyzed and the ENSO state when the peak precipitation occurs most often marks the cell color. If so, why the blue squares do not fit the grid? How have they been obtained? The situation in Fig. 9 is even worse.*

We are sorry for this misunderstanding. The grids in the plots are not the grids of ERA5. The grids which appeared in the first versions od figures were used to avoid interpolating of the mapped values. While mapping precipitation values, the interpolation has a physical sense, in mapping extrema occurrence in bins 1–6 we need to have only integer values, while interpolation does not have any sense. In the new version of figures we made the maps in a different way and the misleading grids did not appear. See the new versions of figures bellow.

*8) Fig. 10. What I said for Fig. 5 panels b) and e) holds here too.*

Here, as in the case of Fig. 5e, the conditional means demonstrated the alternation of weak and strong monsoon years (the influence of the ENSO quasibiennial cycle), and the differences of the adjacent bins represent the amplitude of the quasibiennial cycle in the precipitation which is modulated by the ENSO low-frequency mode.

*9) "Although physical mechanisms explaining the observed cross-scale information transfers are yet to be established, the uncovered causal relations can already be used in machine learning tools for forecasting precipitation anomalies." I think that the first sentence should be expanded. The authors should provide an idea for further investigations of what could be a possible mechanism that makes the low frequency of ENSO impact the monsoon at the 2ys scale, referring to the literature that surely has already investigated the link between ENSO and China monsoon from another angle. The second sentence is very vague and does not provide any valuable information.*

This sentence should not be taken out of context of the concluding paragraph, which we would like to extend by other relevant references as:

The fact that complex evolution of climate, atmosphere, or circulation regimes is influenced by interactions of dynamics on multiple time scales is known (Muñoz et al., 2017; Zhang et al., 2023). For instance, Muñoz et al. (2015) suggest that cross-time scale interactions between different climate drivers improves the predictive skill of extreme precipitation. Hsu et al. (2023) show that multiscale interactions, in particular, scale interactions between the monsoon mean field, two modes of intraseasonal oscillation, and synoptic disturbances, were driving the devastating floods in Henan Province, China during July 2021. Liu et al. (2023) used the multiscale window transform (MWT) and MWT-based energy and vorticity analysis (MS-EVA), to identify

three scales fields: basic-flow fields (>64 days), intraseasonal oscillation fields (8–64 days) and synoptic-scale-eddy fields (<8 days), responsible for the torrential rainfall event, which hit Zhengzhou on July 20, 2021. Ungerovich et al. (2023) emphasize the role of the large scale circulation anomalies associated with ENSO teleconnections in simulation of extreme rainfall events in Uruguay, while Pineda et al. (2023) suggest that the early onset of heavy rainfall on the northern coast of Ecuador in the aftermath of El Niño 2015/2016 was favored by the convective environment in late January due to cross-time-scale interference of the very strong El Niño event and a strong and persistent Madden-Julian oscillation. The presented research, however, is a first step in developing a methodology able to establish a solid statistical evidence for existence of cross-scale causal interactions and to estimate their effect in measurable, physical quantities. In particular, the results presented here can open a new direction in understanding and predicting precipitation anomalies in eastern Asia. Although physical mechanisms explaining the observed cross-scale information transfers are yet to be established, the uncovered causal relations can already be used in statistical or machine learning tools for forecasting precipitation anomalies. In related considerations, Muñoz et al. (2023) propose to find "Windows of Opportunity" in forecasts across timescales by combining wavelet spectral analysis and a non-stationary time-frequency causality analysis. Materia et al. (2024) try to understand the causal factors behind these windows of opportunity using Liang-Kleeman information flow (Liang, 2013). This study demonstrates the ability to identify sources of cross-scale predictability by using complex continuous wavelet transform and information-theoretic approach to causality (Paluš, 2014).

*10) If the QB mode is the one that impacts the dataset the most why, to make all figures up to Fig 5, it was used the 6ys mode instead?*

We report that the effects of both QB and LF modes are important. Using the six bins of the LF phase, both effects are apparent, as described in the paper. Focusing on the QB mode alone, one would effectively average all even and odd bins and the LF mode would attenuate the effect of the QB mode. The present methodology estimates the combined effect of both modes. In principle we could distinguish, in each grid point, which mode has a stronger effect. We are afraid, however, that the manuscript is already very complex to understand, so that we leave such distinction for further research.

*Minor:*
*The sentence "other recent follow-up studies were particularly concerned with the predictability and future of ENSO projections attributable to global to regional scale interconnections, including the combined influence of ENSO and PDO" is unclear.*

Reformulation: Other recent studies considered combined influence of ENSO and NAO or PDO and future ENSO projections.

*"The Yangtze River is China's longest river and the world's third largest, contributes considerably to China's equitable economic and ecological growth", I understand economic growth but what does ecological growth mean?*

The ecological growth of the Yangtze River Basin refers to the development and changes in its ecosystems over time, influenced by both natural processes and human activities. The Yangtze River Basin has undergone significant ecological changes due to various factors such as climate, geography, and human impact. The basin's high, middle, and lower portions have various climates and geomorphology, which contribute to its great biodiversity and huge number of uncommon and unique species (Chen, 2020). Therefore, its ecological growth is as important as economic growth.

*"The YZRB is predominantly controlled by Siberian northwest winter and southeast summer monsoon." Written like this it suggests the existence of a Siberian monsoon. Perhaps the authors are mentioning the Siberian High?*

The YZRB is predominantly controlled by Siberian northwest winter and southeast summer monsoon. This monsoon brings cold, dry air from Siberia during the winter months. It has the potential to diminish temperatures and precipitation, resulting in drought conditions in certain areas of the basin (Yang et al., 2023).

*Fig. 2 carries almost no information. In the text, a lot of geographical details are given "hydrological station (YHS) separates the Yangtze River into upper and lower sections and is renowned as the 'Gateway to the Three Gorges'. The Three Gorges Dam (TGD) lies just approximately 40 kilometers above (Xu et al., 2007). The territory above Yichang station is commonly regarded as the upper sub-basin of the YZRB; the region from Yichang station and Hukou station is the middle sub-basin; and the region under Hukou station is the lower sub-basin of the YZRB (Fang et al., 2018). The YZRB lies in subtropical and temperate climate zone dominated by monsoonal winds; the southern region exhibits subtropical climate while northern region presents temperate zone. Major flooding in YZRB is linked with warm ENSO and strong summer monsoons typically occur after El Niño conditions in the winter, while weak winter" which are impossible to locate in the maps, making both the map and this paragraph useless.*

Thank you for raising this critical issue with Fig. 2. The Referee #2 was more concerned about the lack of an orographic height color bar, and also proposed including the location of the local precipitation stations mentioned in the paper. As a result, we updated the orographic color bar height and mentioned the position of the color bar of local precipitation stations, as seen in the following new version of Fig.2

We will remove the unnecessary information as per your suggestion.

[Figure]

**Figure 2. Study area.** (Top) Localization of the selected region in Yangtze and Yellow River Basins. (Bottom) Study area in a detailed view, including the positions of selected stations.

*Fig 3. To use red for precipitation is a semiotically unfortunate choice. Moreover, one cannot see the rivers.*

We have changed the color for the rivers. We understand that semiotically better choice for precipitation would be a blue color scale. We have remade relevant figures in a blue scale, however, it seems that the red-yellow scale is better readable. We decided to present both versions of the figures and ask referees and editors to advice which to use.

[Figure]

**Figure 3. Precipitation and its variability in the study area.** Spatial distribution of precipitation and its variability during 1951-2020; (a) mean precipitation, (b) precipitation standard deviation (SD), (c) relative difference between ENSO positive and ENSO neutral state, (d) relative precipitation SD (SD/mean precipitation).

*Eq.1 Exp parenthesis is too small*

The equation

$$\psi(t) = \frac{1}{\sqrt{2\pi\sigma_t^2}} \exp(-\frac{t^2}{2\sigma_t^2}) \exp(2\pi i f), \qquad 1$$

will be changed to

$$\psi(t) = \frac{1}{\sqrt{2\pi\sigma_t^2}} \exp\left(-\frac{t^2}{2\sigma_t^2}\right) \exp\left(2\pi i f t\right), \qquad 1$$

*The sentence "The causal delay can be found in the causality analysis" in that part of the text is particularly unclear.*

The sentence "The causal delay can be found in the causality analysis." will be extended as "The causal delay can be found in the causality analysis as follows:"

The text which followed this sentence explains the finding of causal delay: "In Fig. 5a the conditional mutual information represents the causal influence of ENSO states on the precipitation (EASMI-ZQY index). .... the influence of ENSO exhibits a significant peak for the time lag of 6 months."

*Nino3.4 sometimes is written with (often in the text) and sometimes not (e.g. label of fig.4). The authors should be consistent: either with or without. Sometimes, it is written with a coma ("Nino3,4") the point should be employed instead.*

We will unify all occurrences using Niño3.4.

*In Fig5. The panels would benefit of some labels.*

A label for causal directions added, see below.

*Fig. 6 Time unit of measure should be years, plural, not year. The colorbar needs a label*

Modified as recommended, see below.

*Fig 7. "a)", "b)", "c)" and "d)" are barely visible as the rivers and their names. The crosses hide the information about the difference patterns because they are too heavy.*

See new versions below.

*Fig. 8. And 9. Using the same color scale for maximum and minimum precipitation makes very hard to see any differences between the two maps (especially in 8), aside the mean level. Again, rivers colors are unfortunate. "c)" and "d)" labels are barely visible. In fig 9 the color scale of the legend of c and d should be periodic, as the numbers from 1 to 6 represent phases of the 6ys mode, so close numbers should have "close" colors and 1 and 6 should be regarded as close among them.*

See new version below.

*Fig. 10 Labels are oversized compared to panels, a line break or a smaller character dimension should be used.*

See the new version of the Fig. 10.

*In the sentence "for example, in neuroscience, where cross-frequency phase–amplitude coupling has recently been observed in electrophysiological signals reflecting the brain dynamics", I think that mentioning neurosciences here is really out of con-*

[Figure]

**Figure 5. Causal mechanisms and their effects.** (a) Conditional mutual information measuring the causal influence of ENSO states on precipitation characterized by the EASMI-ZQY index (solid blue line) and causality in the opposite direction (dashed black line). The red line is the significance threshold given as the mean+2SD for the surrogate data. (b) Conditional means for the precipitation data from the gridpoint 33.75°N 115.75°E for different ENSO states (ENSO- light-blue, ENSO0 white, ENSO+ light-red) computed for the lag of 6 months. Differences of the adjacent states in black. (c) Evaluation of statistical significance of the maximum relative difference between states, here ENSO- and ENSO0 (red vertical line) using the histogram for the surrogate data (black), (d) Conditional mutual information measuring the causal influence of ENSO 6yr cycle phase on 2yr cycle amplitude for precipitation characterized by the EASMI-ZQY index (solid blue line) and causality in the opposite direction (dashed black line). The red line is the significance threshold given as the mean+2SD for the surrogate data. (e) Conditional means for the precipitation data from the gridpoint 33.75°N 115.75°E for the 6 phase bins within the ENSO 6yr cycle (various colors). Differences of adjacent bins (red) considered as the amplitude of the precipitation quasibiennial cycle. The effect of the 6yr cycle phase is estimated as the maximum difference of the bin values – here the difference between the values of the 6th (orange) and the 3rd (yellow) bins. This value relative to the total precipitation mean is 0.264 and is marked by red vertical line in (f) and found statistically significant in comparison with the surrogate histogram (black).

*text given the journal.*

There are surprisingly many mathematical and data processing methods which have found applications in Earth sciences, neuroscience and other research areas. Besides cross-scale causality and general causality inference methods, we can name various forms of principal or independent component analysis and other dimension reduction methods, complex networks and

[Figure]

**Figure 6. Cross-scale ENSO influence on precipitation in eastern China.** Cross-scale phase-amplitude information transfer characterizing the causal influence of the phase of ENSO quasioscillatory components, with periods given on the abscissa, on the amplitude of precipitation quasioscillatory components with periods given on the ordinate. Significant causal influence of ENSO detected in (a) EASMI-ZQY index, (b) precipitation data from 6 stations from Hu Bei, Jiang Xi and Zhejiang provinces (averaged results), (c) precipitation data from station 58457 Hangzhou from Zhejiang province, (d) precipitation data from station 58527 Jingdezhen from Jiangxi province, (e) precipitation data from 58102 Bozhou station from An Hui province, (f) ERA 5 reanalysis precipitation data from the gridpoint $33.75°N$ $115.75°E$. The colour codes present the conditional mutual information $Z$-score for $Z > 2$, obtained in the test using 100 realizations of surrogate data.

graph-theoretical approaches. For readers seeking a deeper understanding of the method, such an excursion into methodological papers in neuroscience can be useful and inspirative.

We would like to thank again the reviewer for the suggestions which helped us to improve the manuscript.

[revised manuscript text omitted]

---

## Author Response (AR2)

Dear editors and reviewers,

We are submitting the revision of our manuscript

*Cross-scale causal information flow from El Niño Southern Oscillation to precipitation in eastern China.*

In this, hopefully, final revision, in response to the advice of the editor we modified the precipitation maps in Figs. 2, 6, 7 and 8 using the blue scale and marked the rivers and see by the black colour. Also, we have performed minor amendments with editorial support to weed out minor unnecessary segments of text, further polishing the manuscript.

We would like to thank the editor and reviewers for many useful comments and suggestions which, we believe, helped to improve the manuscript and made it suitable for the readerships of Earth System Dynamics.

Yours sincerely,

Milan Palus, for the authors' team

Dr. Milan Palus
Institute of Computer Science CAS
Pod Vodarenskou vezi 2
182 00 Prague 8
Czech Republic
mp@cs.cas.cz